# A tissue engineering approach for repairing craniofacial volumetric muscle loss in a sheep following a 2, 4, and 6-month recovery

**Brittany L. Rodriguez**[1], **Emmanuel E. Vega-Soto**[2], **Christopher S. Kennedy**[2], **Matthew H. Nguyen**[2], **Paul S. Cederna**[1,3], **Lisa M. Larkin**[1,2]*

**1** Department of Biomedical Engineering, University of Michigan, Ann Arbor, Michigan, United States of America, **2** Department of Molecular and Integrative Physiology, University of Michigan, Ann Arbor, Michigan, United States of America, **3** Department of Plastic Surgery, University of Michigan, Ann Arbor, Michigan, United States of America

* llarkin@umich.edu

**Data Availability Statement:** All relevant data are included within figures and the main text of the article.

## Abstract

Volumetric muscle loss (VML) is the loss of skeletal muscle that results in significant and persistent impairment of function. The unique characteristics of craniofacial muscle compared trunk and limb skeletal muscle, including differences in gene expression, satellite cell phenotype, and regenerative capacity, suggest that VML injuries may affect craniofacial muscle more severely. However, despite these notable differences, there are currently no animal models of craniofacial VML. In a previous sheep hindlimb VML study, we showed that our lab's tissue engineered skeletal muscle units (SMUs) were able to restore muscle force production to a level that was statistically indistinguishable from the uninjured contralateral muscle. Thus, the goals of this study were to: 1) develop a model of craniofacial VML in a large animal model and 2) to evaluate the efficacy of our SMUs in repairing a 30% VML in the ovine zygomaticus major muscle. Overall, there was no significant difference in functional recovery between the SMU-treated group and the unrepaired control. Despite the use of the same injury and repair model used in our previous study, results showed differences in pathophysiology between craniofacial and hindlimb VML. Specifically, the craniofacial model was affected by concomitant denervation and ischemia injuries that were not exhibited in the hindlimb model. While clinically realistic, the additional ischemia and denervation likely created an injury that was too severe for our SMUs to repair. This study highlights the importance of balancing the use of a clinically realistic model while also maintaining control over variables related to the severity of the injury. These variables include the volume of muscle removed, the location of the VML injury, and the geometry of the injury, as these affect both the muscle's ability to self-regenerate as well as the probability of success of the treatment.

**Funding:** This work was funded by the National Institutes of Health, National Institute for Arthritis and Musculoskeletal and Skin Diseases (https://www.niams.nih.gov/grants-funding) received by LL (Grant number:1R01AR067744-01), as well as NIH P30 support provided by the Michigan Integrative Musculoskeletal Health Core Center. The funders had no role in study design, data collection and analysis, decision to publish, or preparation of the manuscript.

**Competing interests:** The authors have declared that no competing interests exist.

## Introduction

Craniofacial disorders are often more complex and manifest differently compared to disorders of the trunk and extremities [1]. These facial disorders are often accompanied by a severe loss of skeletal muscle referred to as volumetric muscle loss. Volumetric muscle loss is the traumatic or surgical loss of 20–30% or more of muscle volume. This loss of muscle exceeds the body's inherent capacity for regeneration and often results in persistent functional deficits and cosmetic deformities [2–4]. Craniofacial disorders including VML often necessitate surgical intervention. In fact, reconstructive maxillofacial surgery is the third most common reconstructive procedure performed in the US with over 200,000 surgeries performed annually [5]. Additionally, craniofacial trauma is common in both military and civilian medicine [6–8]. Soft tissue injuries, which include VML, make up over half of the craniofacial injuries that are sustained by civilians [6] and the majority of craniofacial injuries that are sustained in combat [7]. Furthermore, craniofacial injuries make up over a quarter of all injuries sustained on the battlefield injuries and are often life threatening [7,9].

Compared to extremity VML, craniofacial VML presents additional challenges to repair. Craniofacial muscle contributes to facial expression and requires more complex motion and synchrony than trunk and extremity muscles. Additionally, craniofacial muscle is characterized by differences in both its extracellular matrix (ECM) and its satellite cell populations [10–14]. This complexity in tissue structure likely contributes to the noted reduced regenerative capacity of craniofacial muscle compared to trunk and limb muscle which limits the muscle's ability to self-repair following injury [15]. In addition, the difference in embryonic origin between craniofacial muscles (derived from pharyngeal arch mesoderm) and muscles of the trunk and limbs (derived from somatic mesoderm) leads to differences in the muscle stem cell niche and differences in satellite cell phenotype [12,13,16]. For example, all satellite cell populations express Pax7 but only those derived from trunk and limb muscles additionally express Pax3 [16]. Despite these differences, there are currently no animal models of craniofacial VML; studies to date have solely involved VML of the trunk and extremities [17].

Treatments for craniofacial VML including muscle flaps, fillers, and prostheses have significant limitations [18,19]. Muscle flaps involve the transplantation of healthy autogenic muscle tissue into the defect and are limited by tissue availability and donor site morbidity [18,19]. This approach is also often limited by size and shape mismatches between the muscle flap, generally obtained from the leg, and the defect site. The use of fillers and prostheses, including fat grafts and acellular extracellular matrix scaffolds, can address the size-shape mismatch by injecting or implanting biomaterials into the face to correct contour deformities or to bridge functional gaps in the muscle. However, the use of fillers and prostheses can be limited by suboptimal integration of the material and sustained inflammatory response at the repair site [18]. Significantly, despite this multitude of options, no treatments exist that can fully restore normal sensation, expression, and function of craniofacial muscle following VML [1]. Thus, novel treatments need to be developed to reduce the negative structural and psychological effects of facial VML as well as restore normal mechanical function.

In many facial reconstruction surgeries involving the treatment of VML, the zygomaticus major muscle (ZM) is targeted for repair or replacement. This is because of its role in facial expression and its contribution to psychological wellbeing. The zygomaticus major is a superficial muscle that is responsible for pulling the corners of the mouth superiorly and laterally during facial expression. Thus, reduced function of the ZM limits a person's ability to smile, and the impaired ability to smile has been associated with increased levels of depression [20]. Additionally, VML injuries of the ZM are often accompanied by cosmetic disfigurements which are associated with low self-esteem and increased apprehension about appearance [21]. This lack

of satisfaction with physical appearance has a significant effect on social functioning and emotional wellbeing, including a lower frequency of interpersonal behavior [22], making the repair or replacement of the ZM essential for improving quality of life in patients with facial VML.

In this study, we sought to address the gap in craniofacial VML knowledge by creating a large animal model of facial VML. In our previous VML studies, we used our engineered skeletal muscle units (SMUs) to treat a 30% VML in both large and small animal models [23,24]. These studies demonstrated significant functional recovery of the SMU-treated groups compared to unrepaired negative controls. Building upon the success of these studies, herein we sought to modularly scale our SMUs and test their efficacy in repairing VML in a non-mechanically loaded facial muscle. We have previously demonstrated the effectiveness of the modular approach to scaling engineered tissues in large animal (sheep) models of tendon [25], ligament [26,27], and skeletal muscle [24] repair. Herein, we used SMUs to repair a 30% VML deficit in the ovine zygomaticus major muscle and assessed the structural and functional effects of the repair after a 2-month, 4-month, or 6-month recovery period.

## Methods

### Animal care

All animal care procedures followed The Guide for Care and Use of Laboratory Animals [28], according to a protocol approved by the University of Michigan's Institutional Animal Care & Use Committee (Protocol Number: PRO00008906). In all instances, the sheep were first sedated through the administration of intramuscular xylazine (0.2 mg/kg) and then anesthetized through the administration of intravenous propofol (8 mg/kg) and gaseous isoflurane at concentrations between 2–5% to maintain a deep plane of anesthesia. For survival procedures, the animals were fasted and a fentanyl patch (75 mcg/hr) was administered 24hrs prior to surgery. Additionally, the animals received a subcutaneous dose of carprofen (4 mg/kg) immediately after surgery with a subsequent dose administered 24hrs post-op as supplementary analgesia if needed. Perioperatively, an intravenous dose of cefazolin (20 mg/kg) was administered. The fentanyl patch was removed 48hrs after surgery. The animals were monitored daily for 10–14 days after surgery by University of Michigan veterinary staff. As a part of this daily health monitoring, the animals were monitored for signs of pain and changes in eating habits. Surgical staples were removed 10–14 days after surgery. For terminal procedures, all animals were euthanized through the administration of a lethal dose of sodium pentobarbital (195mg/kg) and subsequent bilateral pneumothorax.

### Muscle biopsy collection

A single 4-month-old female Polypay sheep (Oswalt Farms, Vicksburg, MI) was the sole tissue donor for the fabrication of all engineered tissues in this experiment. The animal was euthanized as described above and the semimembranosus muscles were dissected under aseptic conditions. Biopsies were transported to the laboratory in chilled Dulbecco's phosphate-buffered saline (DPBS) (Gibco, cat. no. 14190–250) supplemented with 2% antibiotic-antimycotic (ABAM) (Gibco, cat. no. 15240–062).

### Cell isolation

Muscle progenitor cells including satellite cells were isolated as described previously [23,24,29–34]. Briefly, muscle biopsies between 3g and 3.5g were sanitized in 70% ethanol and finely minced with a razor blade. The minced muscle was placed under UV light for 5 minutes and subsequently added to a digestion solution composed of 2.3 mg/mL dispase (Thermo

Fisher, cat. no. 17105–041) and 0.3 mg/mL collagenase type IV (Thermo Fisher, cat. no. 17104–019). The mixture was incubated for a total of 2.5 hours at 37˚C with constant agitation. Following enzymatic digestion, the resulting suspension was then filtered through a 100μm mesh filter (Fisher Scientific, cat. no. 22-363-549) followed by filtration through a 40μm mesh filter (Fisher Scientific, cat. no. 22-363-547) and centrifuged. The supernatant was discarded, and the cells were re-suspended in freezing medium (50% Dulbecco's Modified Eagle's Medium (DMEM) (Gibco, cat. no. 0569–010), 40% fetal bovine serum (FBS) (Gibco, cat. no. 10437–028), 10% dimethyl sulfoxide (DMSO) (Sigma-Aldrich, cat. no. D2650), supplemented with 1% ABAM) at a concentration of 5,000,000 cells/mL. The cells were frozen to -80˚C at a rate of -1˚C/minute and subsequently stored in liquid nitrogen until plating.

## Establishment of design parameters

Zygomaticus major samples were taken from n = 7 Polypay wethers (castrated males) that were euthanized as part of an unrelated study. Gross measurements of the muscle were taken and the whole muscle was dissected and weighed. We calculated the 95% confidence interval for the ZM weights and set the SMU weight goal to be 30% of that confidence interval. A set of SMUs was fabricated to establish a 95% confidence interval for SMU weights. Using these values, we calculated the number of single SMU units that would have to be modularly combined to completely fill the VML defect.

## Construct fabrication

Frozen cells were removed from liquid nitrogen and quickly thawed in a 37˚C water bath. The cell suspension was re-suspended in muscle growth medium (MGM) (60% F-12 Kaighn's Modification Nutrient Mixture (F12K; Gibco, cat. no. 21127–022), 24% DMEM, 15% FBS, 2.4 ng/mL basic fibroblast growth factor (bFGF; PeproTech, Rocky Hill, NJ, cat. no. 100-18B), 1% ABAM, supplemented with an additional 10μL/mL of 1μM dexamethasone (DEX; Sigma, cat. No. D4902)) [23,29–32,35] and seeded at a density of 10,000 cells/cm$^2$ onto 150mm tissue culture dishes embedded with stainless steel pins positioned 9cm apart. The culture media was changed for the first time on day 4 followed by a Monday-Wednesday-Friday feeding schedule thereafter. On day 6, the media was switched to muscle differentiation media (MDM) (70% M199 (Gibco, cat. no. 11150–059), 23% DMEM, 6% FBS, 1% ABAM, 10μL/mL 1μM DEX, 1μL/mL insulin–transferrin–selenium-X (ITSX; Sigma, cat. no. I1884), and 0.72μL/mL of 50mM ascorbic acid 2-phosphate (Sigma, cat. no. A8960)) to promote differentiation [23,29–32,35]. Monolayers began to spontaneously delaminate on days 10–12 and were fully rolled up by day 15. On day 15, two SMUs were combined onto one plate and pinned at 7cm to allow them to shrink down in length and fuse together. On day 18 or day 20 of the fabrication process, the constructs were implanted into the recipient sheep. Alternatively, a subset of SMUs was reserved for *in vitro* characterization.

## In vitro assessments of SMUs

A subset of the SMUs, termed "sentinel SMUs", was reserved for *in vitro* characterization including biomechanical testing and histology. Approximately 24–48 hours after 3D formation, contractile properties of the single SMUs (not modularly fused) were measured as described previously [23,29–33,35,36]. Briefly, contractions were elicited through field stimulation with a platinum electrode and measured by an optical force transducer (World Precision Instruments, cat. no. SI-KG7A) secured to one end of the construct. Tetanic forces were elicited using a 1s train of 2.5ms pulses at 800, 900mA, and 1000mA and 90, 100, and 120 Hz and measured using custom LabVIEW 2012 software. Our current stimulator can only elicit a

maximum current of 1000mA. At this current, and due to the size of these large constructs, we were unable to reach maximum tetanic force capabilities of the SMU constructs. Therefore, using this equipment, we maxed out the capabilities of our equipment and were only able to elicit maximum tetanic force measures in two constructs (shown as red points in Fig 2C). So, the maximum forces capabilities of the SMUs are underestimated in this study.

After biomechanical testing, SMUs were coated in Tissue Freezing Medium (Fisher Scientific, cat. no. 15-183-13), frozen in dry ice-chilled isopentane, and subsequently cryosectioned at 10μm. Cryosections of SMUs were stained with hematoxylin and eosin (H&E) and Masson's trichrome (Polysciences Inc., cat. no. 25088–1) to examine morphological characteristics of the SMUs, as well as immunohistochemically stained to identify the presence of myosin heavy chain (mouse monoclonal, DSHB, cat. no. MF-20c, 1:200 dilution), laminin (rabbit polyclonal, Abcam, cat. no. ab7463, 1:200 dilution), desmin (mouse monoclonal, Abcam, cat. no. 6322, 1:200 dilution), and α-smooth muscle actin (rabbit polyclonal, Abcam, cat. no. ab5694, 1:100 dilution) as described previously [23,33–35].

## Surgical implantation

Animals used for the surgical implant procedures were 6-7-month-old Polypay wethers (castrated males) weighing 45-55kg. The animals were divided into two experimental groups: VML only (negative control, n = 15) and VML+SMU (n = 15) (Fig 1). The positive control used in this study was the uninjured contralateral ZM. On the day of surgery, the animals were weighed and then anesthetized as described above. A 10cm transverse incision was made approximately 1cm above the mandibular border of the left side of the face (surgical side). The skin and the platysma were reflected to expose the zygomaticus major muscle and the mandibular branch of the facial nerve. Gross measurements of the muscle were taken and a full-thickness longitudinal portion of the ZM constituting 30% of the total muscle volume was dissected (Fig 1A). This value was calculated by taking length, width, and thickness measurements of the muscle, converting volume to mass using the density of skeletal muscle ($1.06g/cm^3$), and then removing mass until 30% of the total muscle mass was reached.

In both groups, a nerve transfer (neurotization) was performed in which the mandibular branch of the facial nerve was transected and re-routed to site of injury and sutured in with 8–0 prolene suture (Ethicon Inc., cat. no 2775G). The VML Only animals (negative control) received the injury and neurotization without any additional repair; the remaining muscle was not sutured together so as to simulate a true negative control (Fig 1B). In the VML+SMU group, the injury was immediately repaired by suturing a modularly fused SMU (2 individual tissue units) within the defect with 6–0 PDS II suture (Ethicon Inc., cat. no. Z432H) (Fig 1C and 1D). In the same way, a nerve transfer to the SMU was performed. The nerve was dissected into several small nerve branches and sutured in with 8–0 prolene suture. In both groups, the skin was sutured closed with 4–0 PDS II suture and the skin was stapled along the incision. All animals were monitored daily for 10–14 days after surgery and then returned to herd housing. The animals were allowed to recover for either two months (2mo.), four months (4mo.), or six months (6mo.) before explant procedures were conducted (n = 10 animals per time point).

## In situ biomechanical testing

Following the recovery period, animals were weighed and then anesthetized. We conducted *in situ* biomechanical testing of both the contralateral and surgical ZMs using a custom biomechanical testing system. Both the contralateral and surgical ZMs were dissected leaving the proximal origin intact. The distal end of the ZM was secured to a strain gauge force transducer (World Precision Instruments, cat. no. FORT1000) to measure the force of the muscle

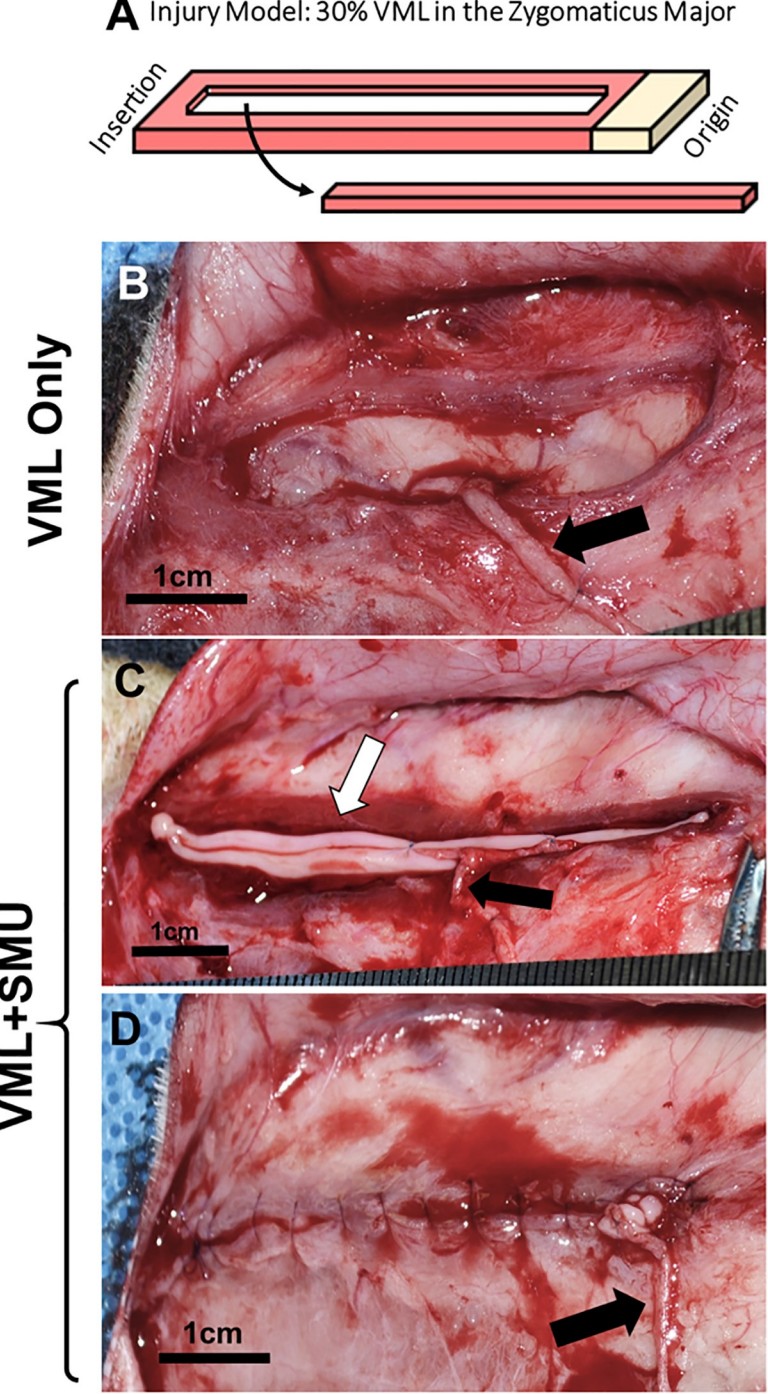

**Fig 1. Experimental groups.** (A) A full-thickness longitudinal portion of the ZM constituting 30% of the total mass was dissected to simulate a VML injury. (B) The VML Only group (negative control) received the injury and nerve re-route without any additional repair. (C) In the VML+SMU group, the injury was immediately repaired by placing n = 2 single SMUs within the defect (white arrow). The re-routed nerve was splayed and sutured to the SMU. (D) The SMU was then sutured into defect site. Black arrows in B-C represent re-routed nerve.

contractions. Contractions were elicited through whole-muscle stimulation using a custom bipolar platinum bezel strip electrode. Biomechanical testing of the muscles was conducted as described previously [23,24,37,38]. Briefly, the muscle was placed in the slack position and single 0.1ms pulses of increasing current amplitudes (i.e. 70, 80, 90, 100, 110, and 120mA) were delivered until peak twitch force was reached. Maintaining the current, the muscle length was subsequently adjusted to the length at which twitch force was maximal. The length of the muscle at which twitch force was maximal was defined as the optimal length (Lo). The stimulus was then switched to a tetanus in which a 600ms train of 0.1ms pulses were delivered. The frequency of these pulses was increased (i.e. 60, 80, 100, 120Hz) until isometric tetanic force was maximal. Data was recorded using custom LabVIEW 2018 software. This process was then repeated on the contralateral (uninjured) ZM muscle. Immediately after biomechanical testing, both the contralateral and surgical ZMs were fully dissected, weighed, and prepared for histology. The animals were subsequently euthanized.

## Histology

After dissection, the muscles were weighed, and gross measurements were taken. The muscles were then divided into segments, coated in tissue freezing medium, and frozen in dry ice-chilled isopentane. Frozen samples were cryosectioned at 10μm and then stained with hematoxylin and eosin (H&E) and Masson's trichrome (Polysciences Inc., cat. no. 25088–1) to examine morphological characteristics of the SMUs. Cross-sectional samples were also immunohistochemically stained to identify myosin heavy chain (mouse monoclonal, DSHB, cat. no. MF-20c, 1:200 dilution), laminin (rabbit polyclonal, Abcam, cat. no. ab7463, 1:200 dilution), perilipin (rabbit polyclonal, Abcam cat. no. ab3526, 1:200 dilution), fast myosin isoform (rabbit polyclonal, Abcam cat. no. ab91506, 1:200 dilution), and slow myosin isoform (mouse monoclonal, Abcam cat. no. ab11083, 1:200 dilution) using a protocol described previously [23,33–35]. Longitudinal samples comprising the entire width of the ZM and ~2cm of the length were sectioned at 25μm and immunohistochemically stained for acetylcholine receptors (α-bungarotoxin, 1:2000 dilution, Life Technologies, cat. no. B1601), synaptic vesicle protein-2 (mouse monoclonal, DSHB, cat. no. SV2c, 1:300 dilution), and neurofilament (mouse monoclonal, BioLegend, cat. no. 837904, 1:1000 dilution) to identify the presence of neuromuscular junctions.

## Cross-sectional area and specific force

A single midbelly cross-section of each ZM was stained for myosin heavy chain (MF20) to identify the presence of skeletal muscle fibers. The total area (cm$^2$) that was positively stained with MF20 was measured using an automated process in ImageJ/Fiji. Because the ZM muscle fibers have no pennation angle [39], the physiological cross-sectional area used to calculate the specific force is equal to the cross-sectional area of the muscle. To calculate the MF20+ specific force, the maximum tetanic force (N) of the muscles was divided by the MF20+ cross-sectional area (cm$^2$) to calculate the specific force (N/cm$^2$) of the muscle. These same images were also used to evaluate the muscle fibers present in the injury site. These fibers were identified based on their location in the tissue rather than their size. The number and cross-sectional area of all of the fibers in the injury site was measured using ImageJ/Fiji in the 6-month recovery group animals. Additionally, total fiber counts were taken for each midbelly ZM cross-section. Using the MF20 and laminin-stained midbelly cross-sections sections, the total muscle fiber count for each muscle was enumerated using ImageJ/Fiji.

   To quantify the total cross-sectional area of the muscle, including both MF20+ muscle fibers as well as connective tissue, nerve, and blood vessels within the muscle, the same MF20

and laminin-stained midbelly ZM cross-sections were used. The total cross-sectional area was measured in ImageJ/Fiji using the freehand selection tool. This total cross-sectional area was then used to calculate the total cross-sectional area specific force (N/cm$^2$) by dividing the maximum tetanic force of the muscle by its total cross-sectional area. The comparison of the MF20 + specific force and the total cross-sectional area specific force gives insight into the change in muscle content, including the increase in fibrotic connective tissue in the injured muscles.

Finally, to quantify the difference in the amount of connective tissue between the uninjured contralateral muscles and the injured ZMs, we calculated the MF20-negative area by subtracting the MF20+ area from the total cross-sectional area and represented it as a percentage of the total cross-sectional area. This portion of the total area that was made up of connective tissue was then normalized to the amount that was present in the contralateral muscle.

### Fiber typing

A single midbelly cross-section of each ZM was immunohistochemically stained for fast myosin isoform to identify the type II muscle fibers and slow myosin isoform to identify type I fibers. To evaluate the percentage of the total muscle cross-sectional area that was type I fibers, the amount of area positively stained for slow myosin isoform was measured using ImageJ/ Fiji. This value was divided by the total area of all muscle fibers which was also quantified using ImageJ/Fiji. To calculate the number of type I fibers relative to the total number of fibers, four regions that were 9.6mm$^2$ in area were chosen at random. In total, these areas constituted ~30% of the total muscle cross-sectional area. The number of fibers expressing slow myosin isoform (type I fibers) and the number of fibers expressing fast myosin isoform (type II fibers) were enumerated. This data was presented as the total number of slow fibers as a percentage of the total number of fibers enumerated. To evaluate the degree of fiber type grouping, we also enumerated the number of grouped fibers in the entire muscle cross-section. A "grouped fiber" is defined as a type I muscle fiber that is completely surrounded by other type I muscle fibers. This data was correlated to the percentage of force recovery for each animal. For these analyses, only animals in the 6-month recovery group were evaluated.

### Intramuscular fat

A single midbelly cross-section of each ZM was also stained for perilipin to identify the presence of adipose within the muscle and within the repair site. The total area that was positively stained with perilipin was measured using ImageJ/Fiji. False-positive staining of the muscle fascia as well as "edge effect" staining was excluded from this analysis. This value was divided by the cross-sectional area of the muscle to calculate the percentage of the area that was positively stained for perilipin.

### Statistical analysis

Statistical analyses were performed using GraphPad Prism software. To evaluate the effects of recovery timepoint and experimental group, values were first normalized to the contralateral muscle and differences were assessed with a two-way ANOVA with Sidak's multiple comparisons test (SMC). If no significant differences between recovery timepoints were present, statistical differences between experimental groups and between the contralateral and surgical muscles were assessed with a two-way repeated measures ANOVA (two-way RM ANOVA) and post-hoc Sidak's multiple comparisons test (SMC). The P-values for interaction and subject were not reported unless significant. Alternatively, a one-way ANOVA and Tukey's multiple comparisons tests (TMC) were used to determine differences between groups. Results were significant at $P<0.05$. Bars on graphs indicate mean ± standard deviation.

# Results

## Establishing SMU design parameters

Native zygomaticus major samples were taken to inform the design parameters for this study and to determine the number of single SMUs that would have to be modularly combined to fill a 30% VML defect. The 95% confidence interval for the total ZM weights of n = 7 animals was 2.0–2.4g. A group of n = 12 SMUs was used to determine the 95% confidence interval for SMU weight which was 0.31–0.39g. Thus, we modularly combined n = 2 SMUs to fill a 30% defect. The modular assembly is depicted in Fig 2A.

## Evaluations of SMUs

Light microscopy of SMU monolayers revealed extensive myotube networking on day 11 just prior to delamination and 3D formation (Fig 2B). We evaluated the histological characteristics of SMUs through staining of both individual SMUs (Fig 2D, 2E and 2H) and modularly combined SMUs (Fig 2F and 2G). 24–48 hours after 3D formation, we evaluated the force production of n = 11 single unit SMUs. The average tetanic force production of a single SMU was $72.0 \pm 42.1\mu N$, with the 95% confidence interval being $43.7–100.3\mu N$ (Fig 2C). However, as described in the methods section, this data is likely an underestimation of the maximum force production of the SMUs, as only n = 2 SMUs peaked below the maximum current and/or frequency allowed by our force testing system (denoted by the red data points in Fig 2C). It should be noted that all sentinel SMUs produced force and contracted following electrical stimulation.

Histological analyses of SMUs revealed muscle fibers present throughout both the single SMU units (Fig 2E) and modular SMUs (Fig 2F) indicating that the modular assembly of the SMUs did not negatively impact the presence of muscle fibers. The SMUs' extracellular matrix was composed of laminin which was identified immunohistochemically (Fig 2E and 2F) and collagen which was identified by the blue Masson's trichrome staining (Fig 2D and 2G). Immunohistochemical staining also revealed the absence of a necrotic core, as DAPI-stained nuclei were present throughout the full thickness of the construct and exhibited normal nuclear morphology (Fig 2I and 2J). In longitudinal sections of a single SMU, we noted the presence of desmin and α-smooth muscle actin arranged linearly, parallel to the longitudinal axis of the SMU (Fig 2H). Positive staining for α-smooth muscle actin without desmin co-staining indicates the presence of myofibroblasts within the SMU.

## Surgical procedures

Across all timepoints, the average percentage of VML was $31.8 \pm 7.1\%$ in the VML Only group and $30.41 \pm 6.1\%$ in the VML+SMU group. The absolute mass of skeletal muscle removed was $0.28 \pm 0.06$g and $0.34 \pm 0.06$g in the VML Only and the VML+SMU groups, respectively. A two-way ANOVA revealed that there were no significant differences in the size of the VML injuries between timepoints (P = 0.5207, n = 10 animals per timepoint) or between experimental groups (P = 0.1424, n = 15 animals per group) (Fig 3A). There were significant differences in the size of the modular SMUs across timepoints (one-way ANOVA: P = 0.0002, n = 5 modular SMUs per timepoint), with the 6mo. animals receiving significantly larger SMUs than the 2mo. (TMC: P = 0.0006) or 4mo. (TMC: P = 0.0003) animals (Fig 3B). On average, the weight of the modular SMUs was $0.41 \pm 0.07$g, $0.35 \pm 0.02$g, and $1.1 \pm 0.35$g for the 2mo., 4mo., and 6mo. groups, respectively. With regards to how much of the VML defect was replaced with an SMU, the defect was filled $106.6 \pm 10.41\%$, $101.7 \pm 4.7\%$, and $169.0 \pm 25.7\%$ in the 2mo., 4mo., and 6mo. groups, respectively (Fig 3C). The modular SMUs in the 6mo. group contributed

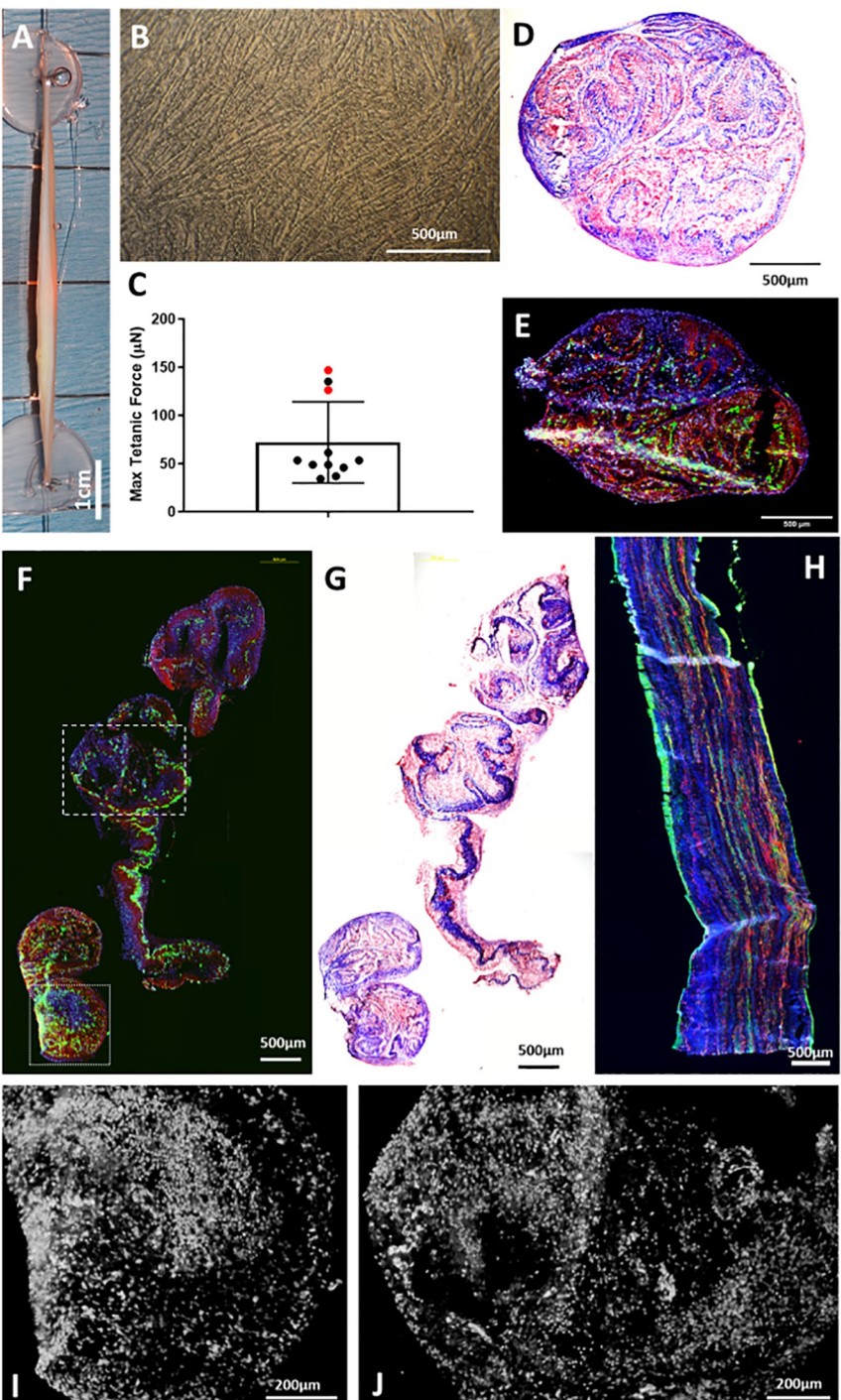

**Fig 2. Characterization of In vitro SMUs.** (A) The final modular assembly of n = 2 SMUs was 7cm long. (B) SMU monolayers prior to delamination showed abundant, networking myotubes. (C) Tetanic forces produced by sentinel SMUs were 72.0 ± 42.1μN on average. Red data points indicate the SMUs whose force peaked below the maximum current and/or frequency allowed by our force testing system. Masson's trichrome staining of a single SMU cross-section (D) and modularly fused SMU cross-section (G) revealed an extracellular matrix composed of collagen. Immunohistochemical staining of a single SMU cross-section (E) and a modularly fused SMU (F) revealed the presence of muscle fibers (MF20, green) and laminin (red), as well as the presence of DAPI-stained nuclei (blue) throughout the thickness of the construct. (H) Immunostaining of a longitudinal section of a single SMU for desmin (red), α-smooth muscle actin (green), and DAPI (blue) revealed a parallel, linear arrangement of these proteins. (I, J) DAPI-stained nuclei (white) of the area denoted by the (I) dotted line box and the (J) hash line box in image (F). Scale bars on B, D-H = 500μm. Scale bars on I, J = 200μm.

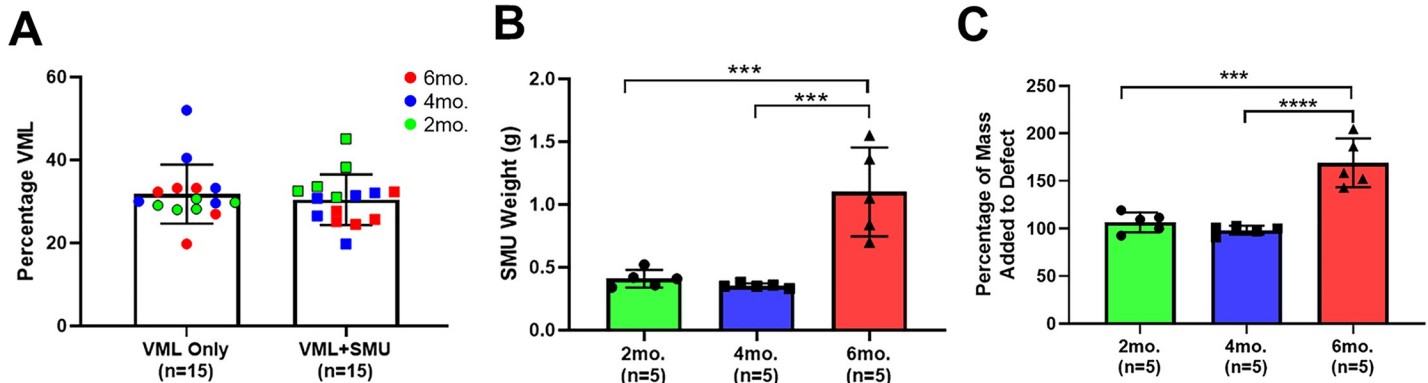

**Fig 3. VML injury and size of SMUs.** (A) The average percentage of the VML injury was 31.8 ± 7.1% in the VML Only group and 30.41 ± 6.1% in the VML+SMU group. There were no significant differences in the size of the VML injuries between timepoints or between experimental groups (two-way ANOVA: P = 0.5207 and P = 0.1424, respectively). (B) The modular SMUs implanted into the 6mo. animals were significantly larger than those in the 2mo. (P = 0.0006) or 4mo. (P = 0.0003) groups and (C) contributed significantly more mass to the defect those in the 2mo. (P = 0.0006) or 4mo. (P = 0.0003) groups. *** indicates P ≤ 0.001; **** indicates P ≤ 0.0001.

significantly more mass to the defect compared to the SMUs in the 2mo. (TMC: P = 0.0001) or 4mo. (TMC: P<0.0001) groups; however, only n = 3 animals had a mass deficit remaining after SMU implantation, and this deficit was 6.5 ± 2.5%.

Although the SMUs implanted into the 6mo. animals were significantly larger than the other SMU cohorts, the fabrication protocol was identical for each cohort of SMUs and the cells used in their fabrication all came from the same donor animal. The difference in SMU size was due to faster cell growth such that after the monolayer delaminated to form an SMU, a subsequent monolayer grew and delaminated around the existing SMU by the time of implantation. Because the 6mo. cohort of SMUs was fabricated first, the cells spent the least amount of time frozen prior to SMU fabrication which likely accounts for the difference in the cells' growth rate. It is important to note that although larger, these SMUs did not exhibit any differences in monolayer quality, morphology, or force production compared to the other sentinel SMUs. Thus, these SMUs likely contributed additional ECM protein, connective tissue cells, and myotubes to the repair site.

## Animal health

Animals were awake, alert, and eating within two hours of completing surgery. None of the animals were observed to have abnormal eating habits after surgery nor were they observed to have any change in motion that would suggest compensation in the contralateral ZM. However, n = 9 out of 30 animals did exhibit signs of low-level amounts of pain, specifically evidenced by bruxism. All animals in the 4mo. and 6mo. recovery groups gained weight normally between the time of implantation and the time of explant. A paired two-way ANOVA performed for each recovery timepoint showed that the experimental group did not significantly affect the animals' body weight (P = 0.6224 for the 2mo. group, P = 0.4439 for the 4mo. group, P = 0.3054 for the 6mo. group, n = 10 per group) (Fig 4). This implies that the surgical procedure was not significantly more stressful for one experimental group over another. As could be expected with growing animals, there was a significant difference in body weight between the time of implantation and the time of explant in the 4mo. (two-way RM ANOVA: P = 0.0003) and 6mo. (two-way RM ANOVA: P<0.0001) groups, but not the 2mo. group (two-way RM ANOVA: P = 0.6206). Notably, although the SMUs were allografts derived from female donors, none of the animals exhibited signs of a chronic immune response at the time of

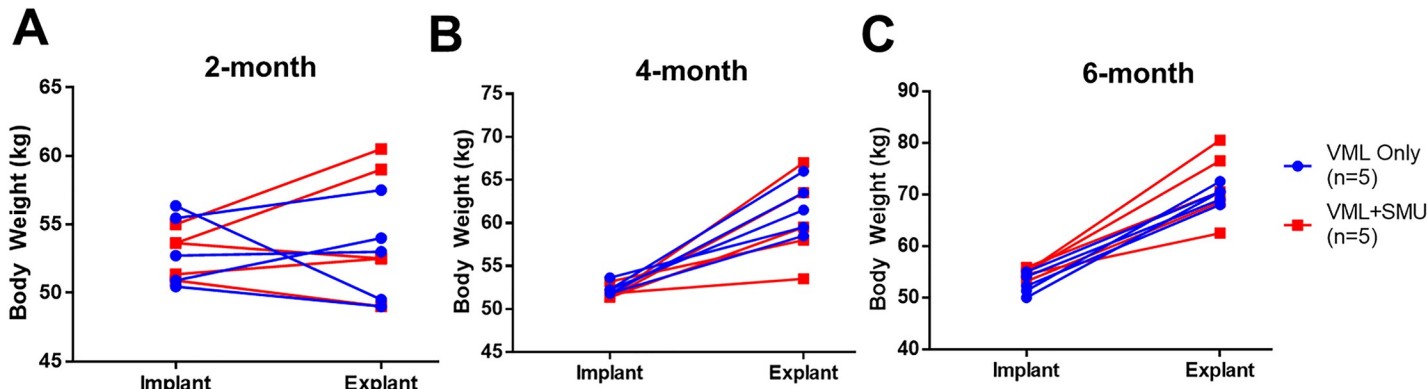

**Fig 4. Animal body weight.** We measured the animals' body weight at the time of implantation and the time of explant for the (A) 2-month, (B) 4-month, and (C) 6-month recovery groups. A paired two-way ANOVA performed for each recovery timepoint showed that the experimental group did not significantly affect the animals' body weight (P = 0.6224 for the 2mo. group, P = 0.4439 for the 4mo. group, P = 0.3054 for the 6mo. group, n = 10 per group). However, there was a significant difference in body weight between the time of implantation and the time of explant in the 4mo. (P = 0.0003) and 6mo. (P<0.0001) groups, but not the 2mo. group (P = 0.6206).

explant (i.e. white blood cell counts were within normal limits). However, the use of allografts may have contributed to an inflammatory response that was not specifically measured in the bloodwork.

## Gross observations at explant

In all animals at the time of explant, there was an abundance of connective tissue surrounding the surgical ZM and tethering it to surrounding tissues. This layer was thickest on the superficial side of the muscle, tethering it to the skin. The tethering was also evident during biomechanical testing. In some instances, we also noted severe atrophy and changes in gross morphology of the ZM muscle. A total of n = 5 animals experienced severe muscle wasting in which histological analyses revealed there were little to no muscle fibers present in the midbelly of the muscle. This was likely the result of ischemia which led to coagulative necrosis [40] and would have been caused by damage to the vasculature supplying the ZM when the initial VML injury was created. Specifically, out of the five animals exhibiting coagulative necrosis, n = 3 were part of the 4-month recovery timepoint (n = 2 VML Only, n = 1 VML+SMU) and n = 2 animals were part of the 2-month recovery timepoint (n = 1 VML Only, n = 1 VML+SMU). These animals were excluded from biomechanical and histological analyses. An additional n = 2 animals were excluded from biomechanical analyses (n = 1 VML Only, 4-month timepoint and n = 1 VML+SMU, 2-month timepoint) because we failed to collect accurate force data for those animals; however, the explanted muscles were still used in histological analyses.

## Mechanical properties of explanted muscles

To evaluate the force capabilities of the injured ZMs, we measured the maximum tetanic force of the surgical ZM and that of the uninjured contralateral ZM (Fig 5A and 5B). We represented the force production of the surgical ZM as a percentage of the force produced by the uninjured contralateral to evaluate the effect of the recovery timepoints and experimental group (Fig 5A). A two-way ANOVA showed there was no significant difference in force between timepoints (P = 0.3576) or between experimental groups (P = 0.3041). After removing 30% of the muscle, one might expect a minimum force production of 70% of the contralateral; however, as is characteristic of VML injuries, the force deficits were much more severe, with

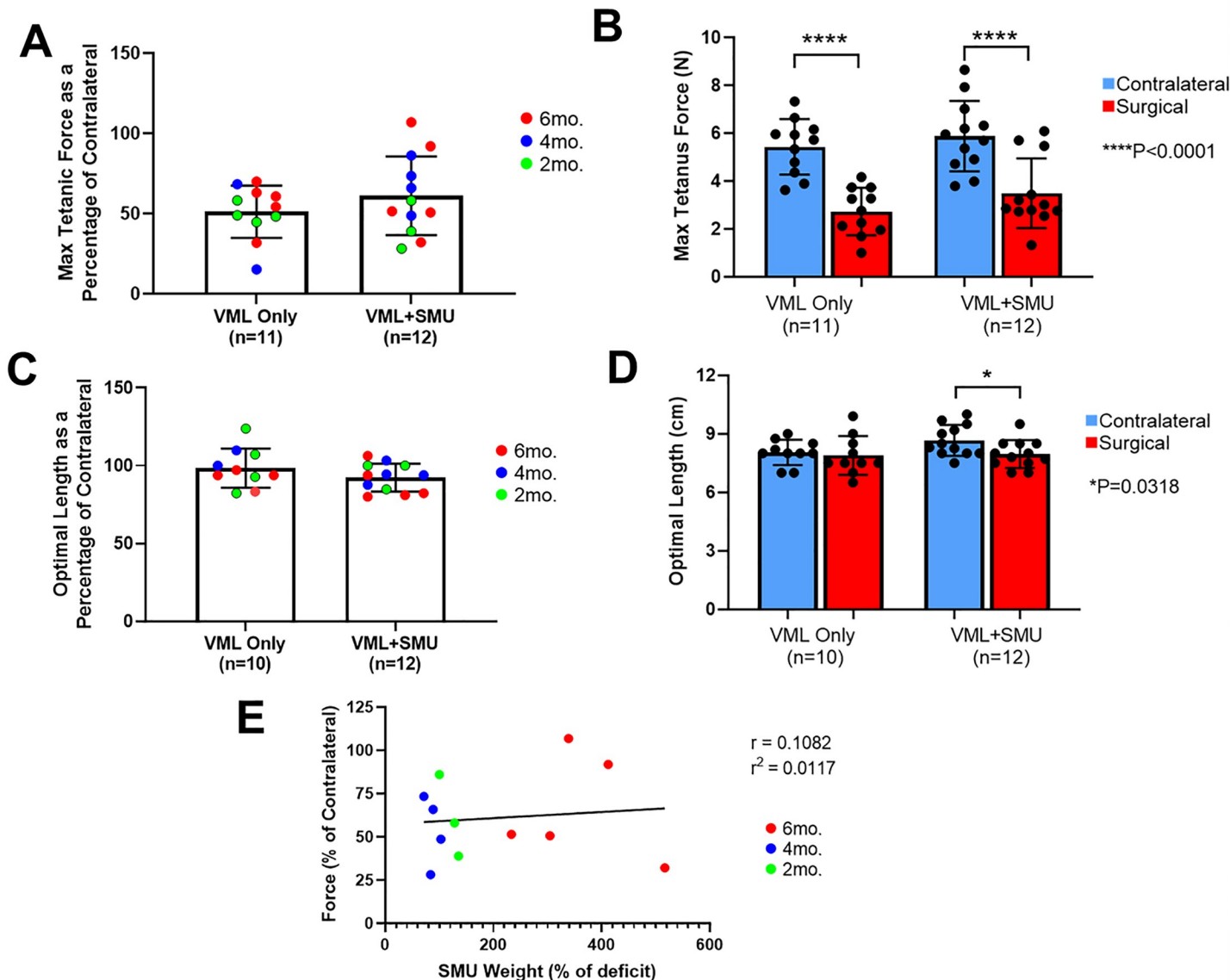

**Fig 5. Mechanical properties of explanted muscles.** (A) A two-way ANOVA showed there was no significant difference in force of the injured ZMs as a percentage of the contralalteral between timepoints (P = 0.3576) or between experimental groups (P = 0.3041). (B) In both experimental groups, the maximum tetanic force of the surgical ZM was significantly lower than the uninjured contralateral (P<0.0001 in both groups). (C) As a percentage of the contralateral, there was no difference in the optimal lengths of the muscle between recovery timepoints (P = 0.2270) or experimental groups (P = 0.1905). (D) There was no siginificant difference in the optimal length of the VML Only group realtive to the contralateral (P = 0.8424), but the optimal length of the VML+SMU group was significantly lower than the contralateral (P = 0.0318). (E) Interestingly, there was no correlation (r = 0.1082) between the SMU weight (as a percentage of the mass deficit) and the percentage of force recovery (P = 0.7379).

the average percentage of force being 51.2 ± 16.3% in the VML Only group (n = 11) and 61.1 ± 24.6% in the VML+SMU group (n = 12). Notably, n = 4 animals in the VML+SMU group exceeded this 70% force capability, while no animals in the VML Only group exceeded this 70% force capability (Fig 5A). Given that the timepoints did not significantly affect the force, we used a paired two-way ANOVA and combined all recovery timepoints for the next statistical analysis. This analysis found that there was a significant difference in the force

capabilities between the surgical and contralateral ZMs (P<0.0001) and no significant differences between experimental groups (P = 0.1830) (Fig 5B). A Sidak's multiple comparisons test showed that in both experimental groups, the maximum tetanic force of the surgical ZM was significantly lower than the uninjured contralateral (P<0.0001 in both groups, n = 11 for VML Only, n = 12 for VML+SMU) (Fig 5B).

Interestingly, the percentage of the defect that the SMUs filled was not correlated with the percentage of force measured (Fig 5E). Although the SMUs in the 6mo. recovery group were significantly larger than the SMUs in the 2mo. or 4mo. groups, the was no correlation between the SMU weight as a percentage of the VML mass deficit and the percentage of force recovery (r = 0.1082, P = 0.7379, n = 12).

To evaluate any changes in the optimal length (Lo) of the muscle fibers within the injured ZMs muscles, we compared the Lo of the surgical and contralateral ZM muscles (Fig 5C and 5D). Because fibers run from origin to insertion in facial muscle, the Lo of the muscle fibers is equal to the optimal length of the whole ZM muscle [39]. We represented the optimal length of the surgical ZM as a percentage of the optimal length of the uninjured contralateral muscle to evaluate the effect of the recovery timepoints and experimental group (Fig 5C). A paired two-way ANOVA revealed that there was no significant difference in optimal length between timepoints (P = 0.2270) or between experimental groups (P = 0.1905). Given there was no significant differences between recovery timepoints, we used a paired two-way ANOVA and combined all recovery timepoints for the next statistical analysis. This analysis found that there was a significant difference in the muscle's optimal length between the surgical and contralateral ZMs (P = 0.0428) and no significant differences between experimental groups (P = 0.2308) (Fig 5D). A Sidak's multiple comparisons test showed that in the VML+SMU group, the optimal length of the surgical ZM was significantly lower than the uninjured contralateral (P = 0.0318, n = 12), but there was no significant difference in the VML Only group (P = 0.8424, n = 10).

Regarding the frequency-force relationship, there were no significant differences in the frequency required to produce a maximum tetanus between injured and the uninjured ZMs. For the uninjured contralateral ZM, maximum tetanus was achieved at 114.0 ± 22.7 Hz. In the injured ZMs, maximum tetanus was achieved at a frequency of 113.6 ± 15.7Hz and 113.3 ± 22.3Hz in the VML Only and VML+SMU groups, respectively. Statistically, there was no significant difference between these groups (one-way ANOVA: P = 0.9956), indicating that the force-frequency relationship of the muscles was unchanged.

## Muscle cross-sectional area and specific force

To evaluate the quantity of skeletal muscle present after the recovery period, we measured the cross-sectional area of the muscle by staining midbelly cross-sections of both the contralateral and surgical ZMs for myosin heavy chain (MF20) (Fig 6A and 6B). We normalized the MF20 + area of the surgical ZM to the contralateral and found that there was no significant difference in normalized MF20+ area between recovery timepoints (P = 0.3462) or between experimental groups (P = 0.7293). The average normalized MF20+ area was 57.4 ± 20.9% in the VML Only group (n = 12) and 54.9 ± 13.4% in the VML+SMU group (n = 13) (Fig 6A). Combining all timepoints, there was no significant difference in total MF20+ area between the experimental groups (two-way RM ANOVA: P = 0.7481), but there was a significant difference between the contralateral and surgical ZMs (two-way RM ANOVA: P<0.0001) (Fig 6B). Using a Sidak's multiple comparisons test, there was a significant difference between the MF20+ area in the contralateral and surgical ZMs of both experimental groups (P<0.0001 for each group; n = 12 for VML Only, n = 13 for VML+SMU) indicating a deficit in the muscle CSA in both groups at the time of explant.

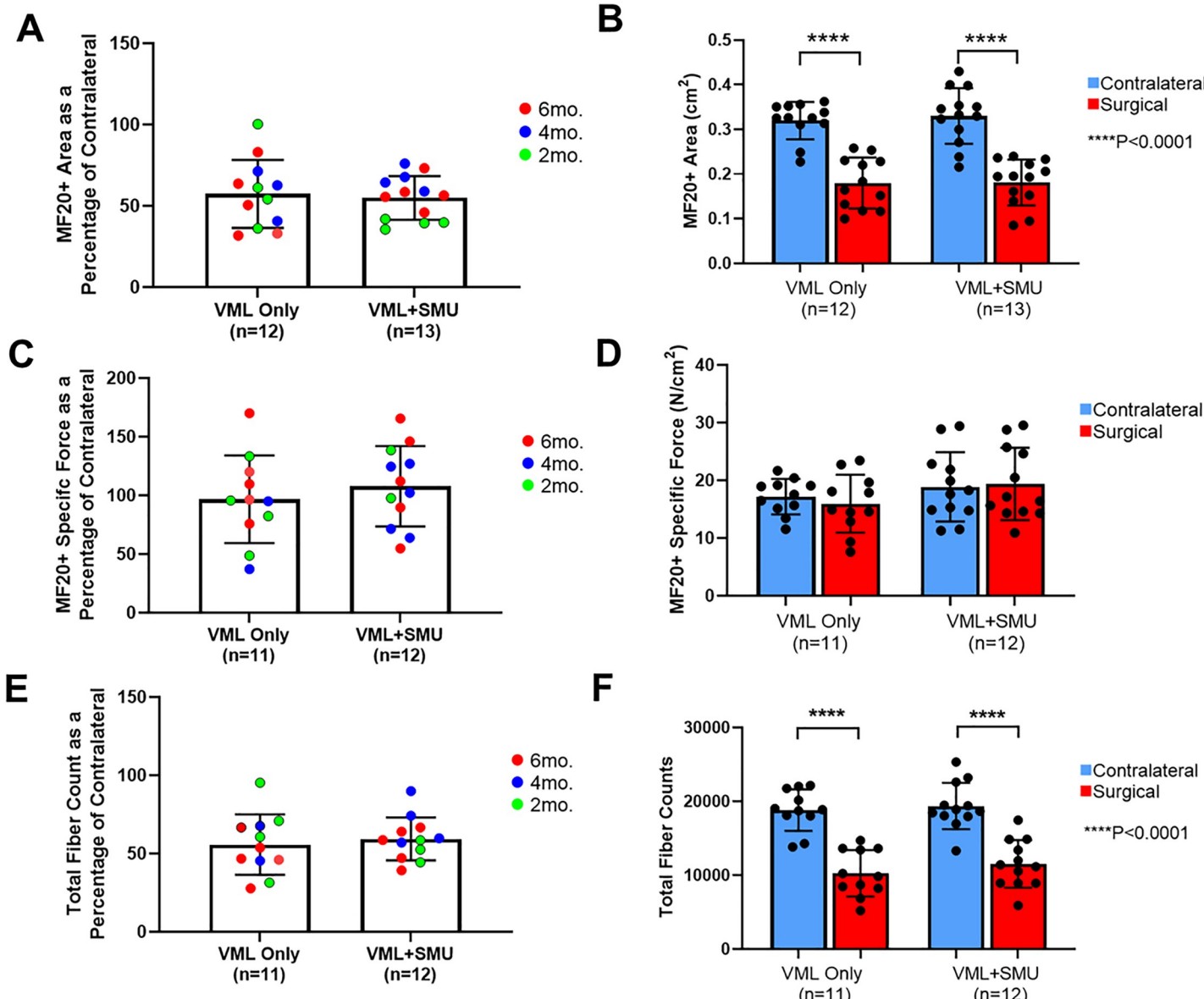

**Fig 6. MF20+ area, specific force, and fiber counts.** (A) A two-way ANOVA showed there was no significant difference in normalized MF20+ area between recovery timepoints (P = 0.3462) or between experimental groups (P = 0.7293). (B) In both experimental groups, the MF20+ area of the surgical ZM was significantly lower than the uninjured contralateral (P<0.0001 in both groups, n = 12 for VML Only, n = 13 for VML+SMU). (C) There was no significant difference in normalized specific force between recovery timepoints (P = 0.2247) or between experimental groups (P = 0.3608). (D) Interestingly, in both experimental groups, there was no significant difference in specific force between the contralateral and surgical ZMs (P = 0.7653, n = 11 for VML Only; P = 0.9529, n = 12 for VML+SMU). (E) A two-way ANOVA showed there was no significant difference in normalized muscle fiber counts between recovery timepoints (P = 0.4046) or between experimental groups (P = 0.7197). (F) In both experimental groups, the total number of muscle fibers in the surgical ZM was significantly lower than the uninjured contralateral (P<0.0001 in both groups, n = 11 for VML Only, n = 12 for VML+SMU).

Because native ZM muscle fibers have no pennation angle [39], the physiological cross-sectional area used to calculate the specific force is equal to the cross-sectional area of the muscle. Thus, the MF20+ cross-sectional area discussed in the previous paragraph was used to calculate the specific force of the ZMs (Fig 6C and 6D). Normalized to the specific force of the contralateral, the mean normalized specific force was 96.8 ± 37.3% in the VML Only group (n = 11) and 107.8 ± 34.3% in the VML+SMU group (n = 12). A two-way ANOVA showed

that there was no significant difference in the specific force as a percentage of the contralateral between timepoints (P = 0.2247) or between experimental groups (P = 0.3608). Given that the timepoints did not significantly affect normalized specific force, we combined all timepoints and found there was no significant difference in the specific force between experimental groups (P = 0.1689) and between the contralateral and surgical ZMs (P = 0.7789) using a paired two-way ANOVA. Notably, there was no significant difference in the specific force between the surgical ZM and the uninjured contralateral ZM in both the VML Only (P = 0.7653, n = 11) and the VML+SMU (P = 0.9529, n = 12) group. Explicitly, the mean specific force of the contralateral, VML Only, and VML+SMU groups were 18.1 ± 4.8N/cm$^2$, 16.0 ± 5.0 N/cm$^2$, and 19.4 ± 6.3 N/cm$^2$, respectively.

We also took a measure of total fiber counts in cross-sections of the ZM midbelly. A two-way ANOVA revealed that there was no significant difference in the total fiber counts as a percentage of the contralateral ZM between recovery timepoints (P = 0.4046) or between experimental groups (P = 0.7197). Specifically, the mean fiber count as a percentage of the contralateral was 55.7 ± 19.3% in the VML Only group and 59.4 ± 13.7% in the VML+SMU group. Given that the timepoints did not significantly affect normalized specific force, we combined all timepoints and found there was no significant difference in the total number of muscle fibers between experimental groups (P = 0.4054) but the total number of muscle fibers was significantly lower in the surgical ZMs compared to the contralateral (P<0.0001) in both experimental groups. This indicates that the loss of MF20+ cross-sectional area in the injured ZMs was consistent with a loss in muscle fiber number.

We calculated the total cross-sectional area of the injured and uninjured ZMs to evaluate whether there were changes in the muscle size as a result of the VML injury. We normalized the total cross-sectional area of the surgical ZM to the contralateral and found that there was no significant difference in normalized MF20+ area between recovery timepoints (P = 0.2258) or between experimental groups (P = 0.6753). The average normalized total cross-sectional area was 74.0 ± 27.2% in the VML Only group (n = 11) and 79.6 ± 15.1% in the VML+SMU group (n = 12) (Fig 7A). Combining all timepoints, there was no significant difference in total cross-sectional area between the experimental groups (two-way RM ANOVA: P = 0.5769), but there was a significant difference between the contralateral and surgical ZMs (two-way RM ANOVA: P<0.0001) (Fig 7B). Using a Sidak's multiple comparisons test, there was a significant difference between the total cross-sectional area in the contralateral and surgical ZMs of both experimental groups (P = 0.0008, n = 11 for VML Only; P = 0.0070, n = 12 for VML +SMU) indicating loss of ZM cross-sectional area in both experimental groups at the time of explant with respect to the uninjured contralateral.

The specific force calculated using the MF20+ cross-sectional area revealed that there was no significant difference between the injured and uninjured ZMs which indicates that the remaining muscle tissue was healthy. However, calculating specific force using the total cross-sectional area of the muscle, which includes the muscle fibers as well as connective tissue, reveals the overall tissue quality of the ZM. Using the total cross-sectional area specific force and normalizing it to the contralateral, the mean normalized total cross-sectional area specific force was 74.8 ± 32.5% in the VML Only group (n = 11) and 77.5 ± 27.6% in the VML+SMU group (n = 12) (Fig 7C). A two-way ANOVA showed that there was no significant difference in the total-cross sectional area specific force as a percentage of the contralateral between experimental groups (P = 0.6753). Combining all timepoints, there was no significant difference in the total cross-sectional area specific force between experimental groups (P = 0.2594) but there was a difference between the contralateral and surgical ZMs (P = 0.0009) using a paired two-way ANOVA. Notably, there was a significant difference in the specific force between the surgical ZM and the uninjured contralateral ZM in both the VML Only

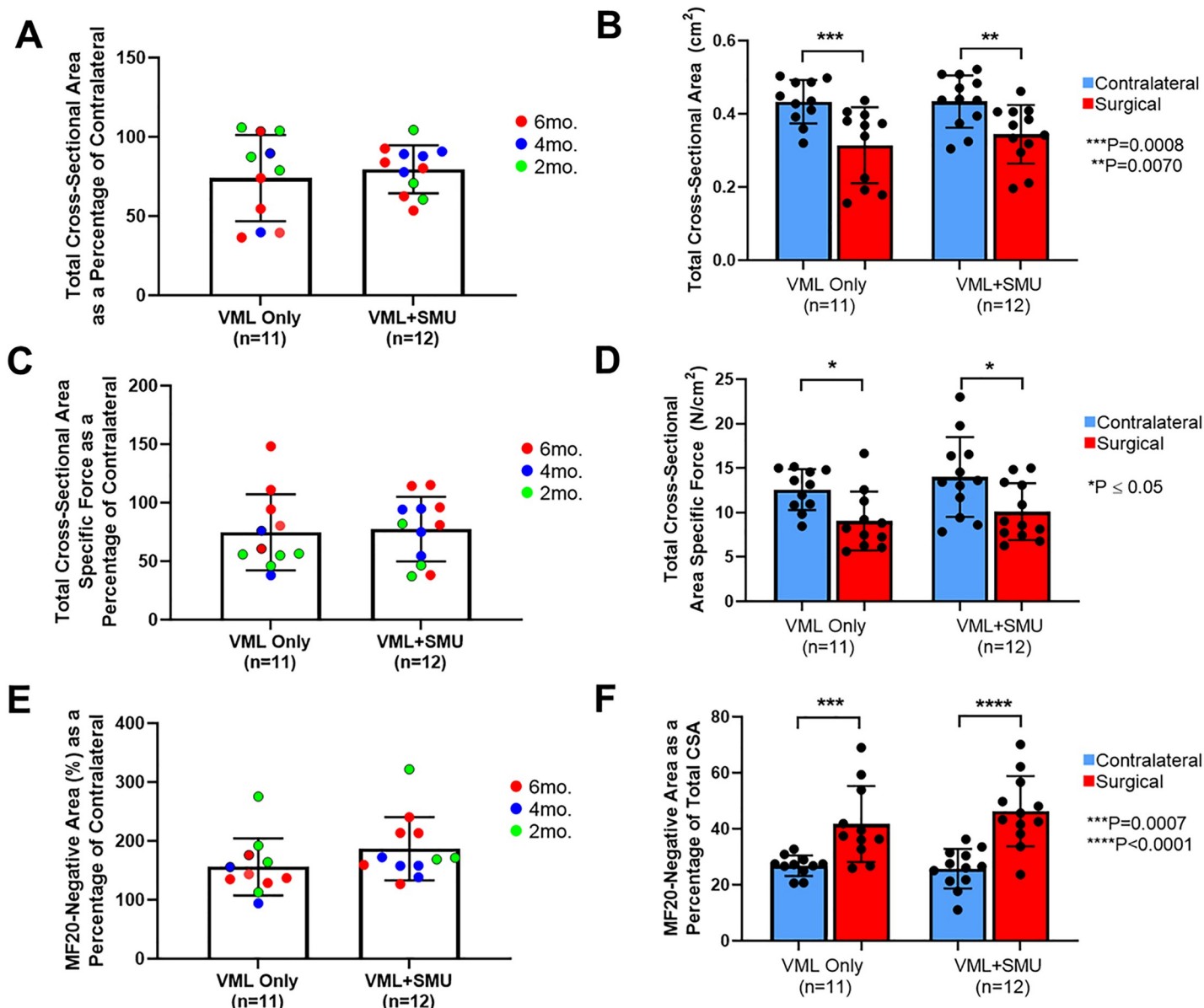

**Fig 7. Total cross-sectional area and specific force.** (A) A two-way ANOVA revealed that there was no significant difference in the total cross-sectional area as a percentage of the contralateral between experimental groups (P = 0.6753). (B) In both experimental groups, the total cross-sectional area of the surgical ZM was significantly lower than the uninjured contralateral (P = 0.0008, n = 11 for VML Only; P = 0.0070, n = 12 for VML+SMU). (C) There was no significant difference in normalized total cross-sectional area specific force between experimental groups (P = 0.6753). (D) In both experimental groups, the total cross-sectional area specific force was significantly lower in the surgical ZM than the uninjured contralateral (P = 0.0384, n = 11 for VML Only; P = 0.0165, n = 12 for VML+SMU). (E) There was no significant difference in the MF20-negative area as a percentage of the contralateral between recovery timepoints (P = 0.1138) or between experimental groups (P = 0.1016). (F) In both experimental groups the percentage of the total cross-sectional area that did not stain for MF20 was significantly higher than the uninjured contralateral (P = 0.0007, n = 11 for VML Only; P<0.0001, n = 12 for VML+SMU).

(P = 0.0384, n = 11) and the VML+SMU (P = 0.0165, n = 12) group (Fig 7D). This indicates that the functional capacity of the ZM as a whole was decreased as a result of the VML injury.

To quantify the amount of connective tissue present in the ZM, that is MF20-negative tissue, we subtracted the MF20+ area from the total cross-sectional area and represented this as a percentage of the total area. When normalizing the percentage of MF20-negative tissue to the control, a two-way ANOVA revealed that there was no significant difference between recovery

timepoints (P = 0.1138) or between experimental groups (P = 0.1016). Specifically, the MF20-negative area was 156 ± 48.4% of the contralateral in the VML Only group and 187 ± 53.7% of the contralateral in the VML+SMU group, indicating an over 50% increase in connective tissue in the injured ZMs (Fig 7E). Combining all recovery timepoints, there was no significant difference between experimental groups (P = 0.6133) but there was a significant difference between the contralateral and surgical ZMs (P<0.0001). Specifically, the percentage of the cross-sectional area that was MF20-negative was 26.3 ± 5.5%, 41.7 ± 12.9%, 46.3 ± 12.0% in the contralateral, VML Only, and VML+SMU groups, respectively (Fig 7F). Overall, this indicates a significant increase in the connective tissue content of the injured ZMs of both experimental groups at all recovery timepoints.

### Histology of explanted muscles

We performed qualitative and quantitative histological analyses on midbelly ZM cross-sections of both the contralateral and surgical ZMs. Fig 8 depicts representative histology of animals in the 6mo. timepoint; we did not qualitatively observe differences in the histology across time-points. In all surgical groups and at all timepoints, the injury site was characterized by a fibrotic region, as noted by the H&E staining (Fig 8A–8C) and the Masson's trichrome stain-ing (Fig 8D–8F). The blue regions in Masson's trichrome-stained sections depict the collagen deposition in the injury site. Immunostaining for myosin heavy chain (MF20) and laminin (Fig 8G–8I) showed that there were small muscle fibers and laminin in the injury site. The amount of MF20 staining was quantified and is depicted in Fig 6.

We noted the presence of muscle fibers of various sizes in the injury site of all experimental groups and all recovery timepoints (Fig 9). Using the MF20 and laminin-stained sections, we quantified the number and size of these fibers in the injury site of animals in the 6mo. recovery

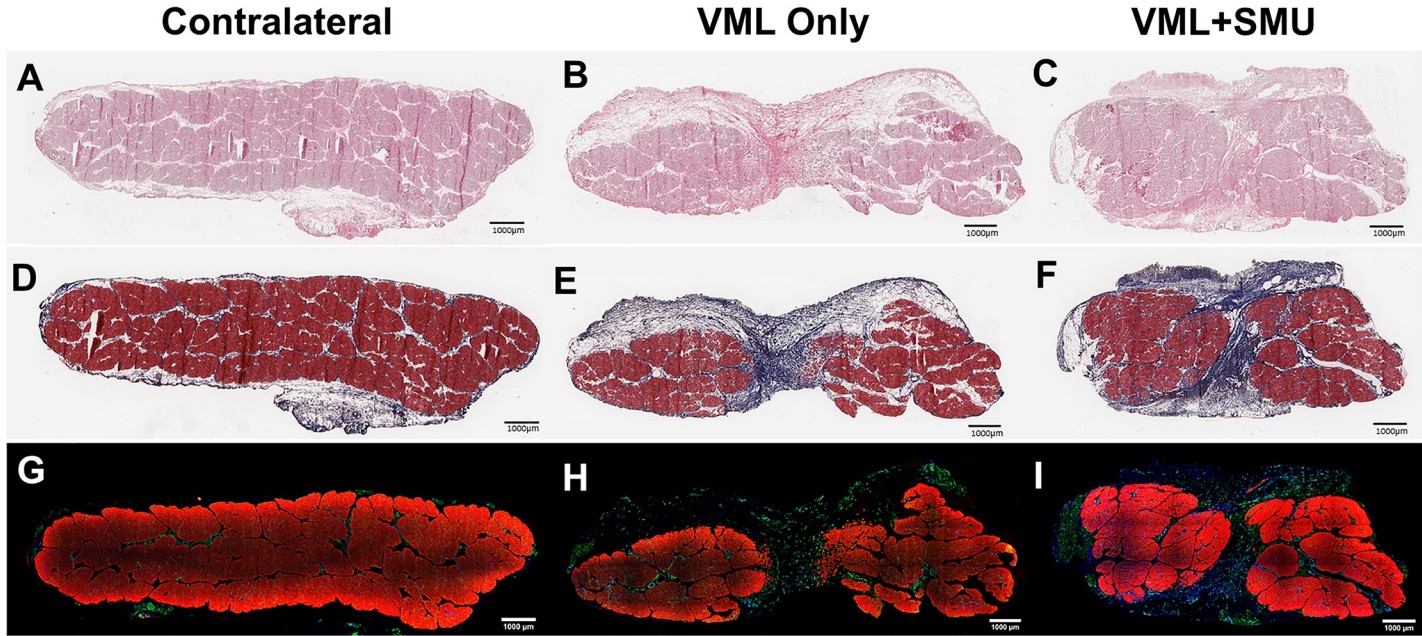

**Fig 8. Histology at the 6-month recovery timepoint.** Cross sections of explants from contralateral (A,D,G), VML only (B,E,H), and VML+SMU (C,F,I) groups were taken from the midbelly of the muscle. Tissues stained with H&E (A-C) show that the surgical site is characterized by the presence of disorganized, hypercellular tissue. Serial sections stained with Masson's trichrome (D-F) indicates that there are large fibrotic regions in the injury site, evidenced by positive collagen staining (blue) that spans between the native muscle (red) in the surgical groups (E,F). Immunostaining for myosin heavy chain (MF20, red) and laminin (green) (G-I) show that muscle is absent in a large portion of the surgical site.

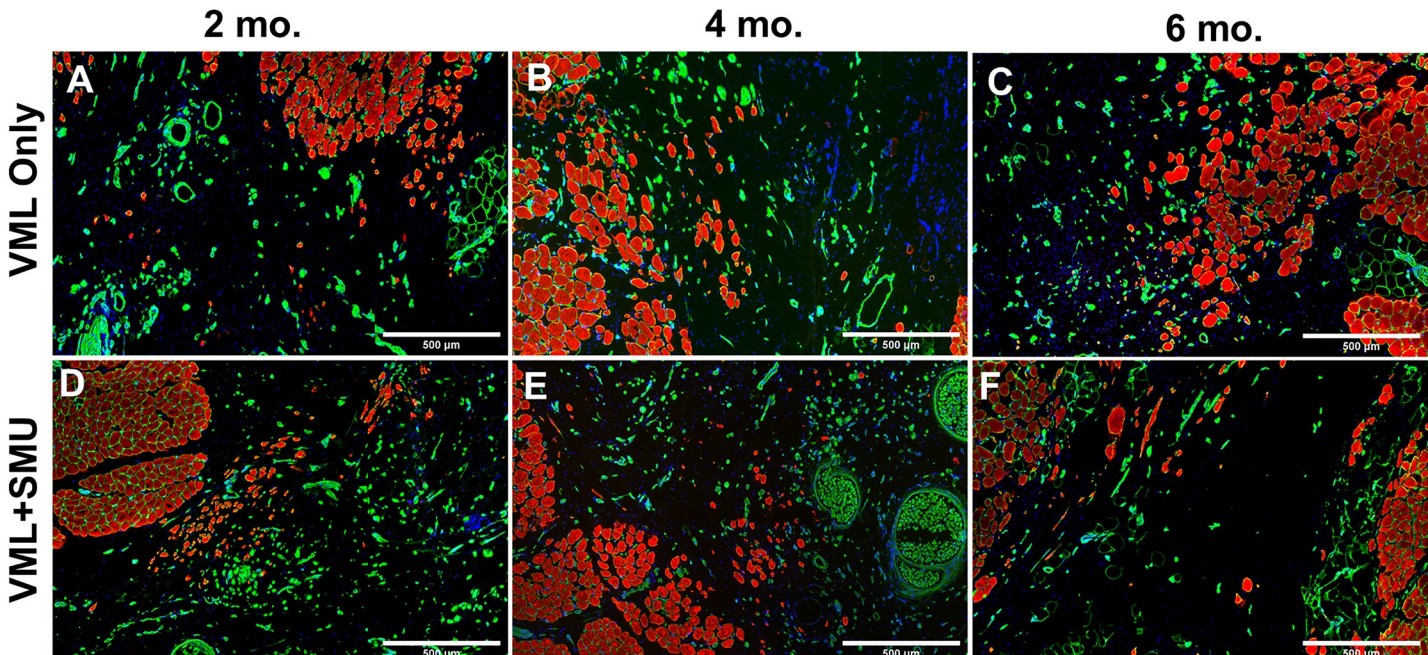

**Fig 9. Muscle fibers in the injury site.** Immunohistochemical staining for myosin heavy chain (MF20, red), laminin (green), and DAPI (blue) revealed the presence of small muscle fibers within the injury site in both the VML only (A-C) and VML+SMU (D-F) groups. These smaller fibers were noted at the 2-month (A,D), 4-month (B, E), and 6-month (C,F) recovery timepoints.

group. There were 58 ± 49 fibers in the injury site of animals in the VML Only group versus 183 ± 197 fibers in the VML+SMU group; however, these values were not significantly different (P = 0.1407, n = 5 for VML Only, n = 4 for VML+SMU). The cross-sectional area of these regenerated fibers was 672.1 ± 691.7μm$^2$ in the VML Only group and 560.0 ± 7507.1μm$^2$ in the VML+SMU group with the VML Only group having significantly larger fibers in the injury site than the VML+SMU group (P = 0.0029, n = 289 fibers for VML Only, n = 916 fibers for VML+SMU). Although the VML Only group's fibers were larger, the origin of these fibers is unknown and these fibers could be from disrupted fascicles, rather than regenerating muscle. At the 6mo. timepoint, we also noted the presence of central nuclei in 80% of the animals in the VML only group (n = 4/5 animals) and 100% of the animals in the VML+SMU group (n = 5/5 animals) indicating that these muscle fibers were still regenerating even 6months after the initial injury.

We also immunohistochemically stained for the presence of neuromuscular junctions (NMJs) in longitudinal sections of the injury site and in the uninjured contralateral muscles (Fig 10). An NMJ was noted to be present if there was an overlap of the pre-synaptic and post-synaptic structures which resulted in a yellow region due to the overlap of the red and green fluorophores. This overlapping of structures was observed in the contralateral ZM at all recovery timepoints, and at least one NMJ was identified in the repair site of both the VML Only and the VML+SMU groups at the 6mo. timepoint. However, NMJs were widely absent in the majority of the repair site of these experimental groups and the images shown in Fig 10 depict what were usually the only NMJs present. Additionally, although acetyl choline receptors and neurofilament were present at earlier timepoints, small muscle fibers in the injury site do not appear to be reinnervated (no overlapping structures) in the 2mo. and 4mo. recovery groups. Overall, in all experimental groups at all timepoints, there appears to be a lack of innervation in the repair site.

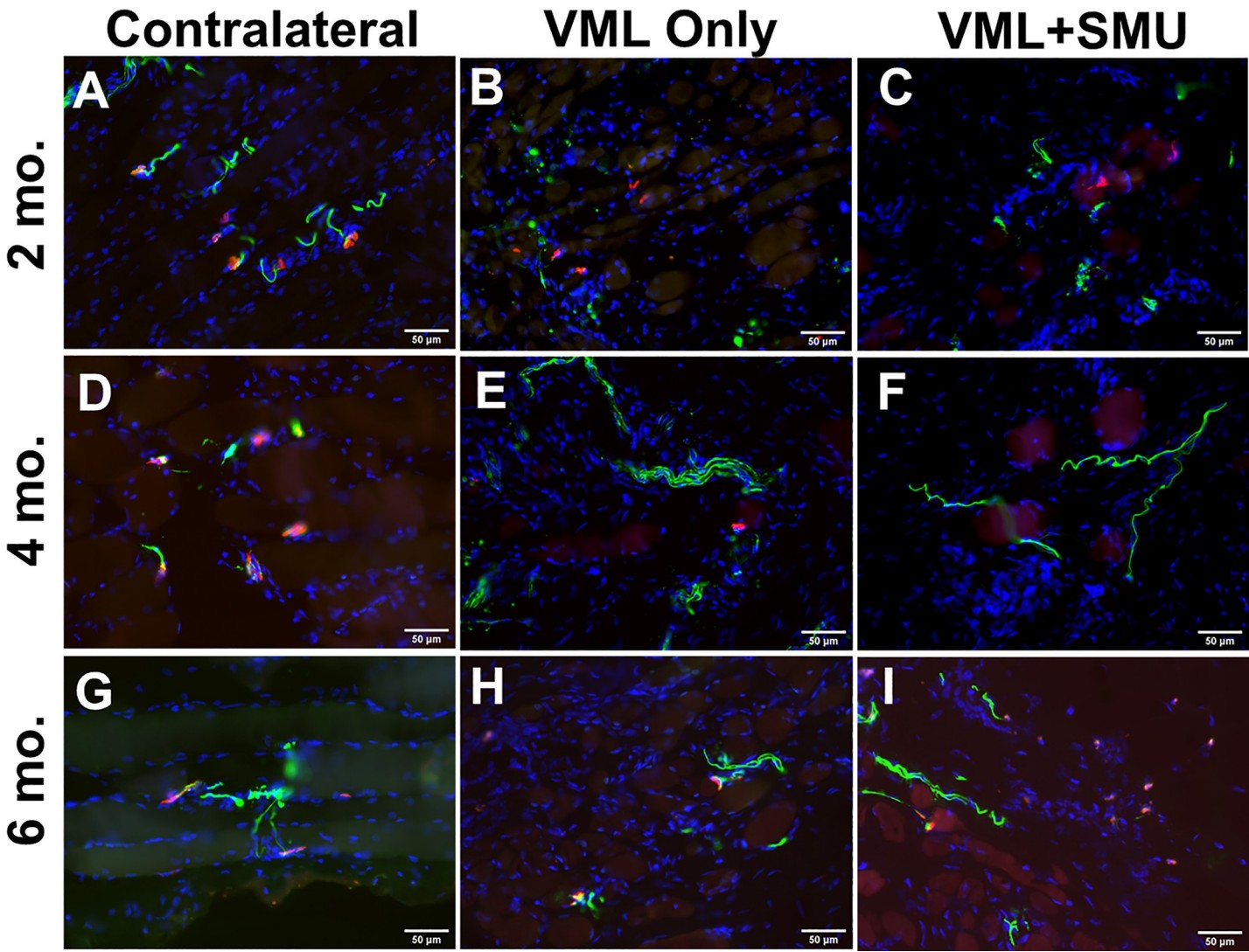

**Fig 10. Neuromuscular junctions in the injury site.** Immunostaining of longitudinal sections for acetylcholine receptors (red), synaptic vesicle protein-2 (green), and neurofilament (green) was performed to identify the presence of neuromuscular junctions. We noted the presence of neuromuscular junctions (i.e. overlap of the pre and post-synaptic structures) in the contralateral muscles (A,D,G) as well as the injury site of the (H) VML Only group and (I) VML+SMU group at the 6mo. timepoint. The small muscle fibers in the injury site do not appear to be fully innervated at the (B-C) 2mo. and (E-F) 4mo. timepoints. Scale bars = 50μm.

We also stained the muscles in the 6mo. recovery group for fast and slow myosin isoforms to see if there was any difference in the fiber type percentage and distribution relative to the contralateral. Histology revealed that there appears to be some difference in the fiber type composition in both the VML Only group and VML+SMU group relative to the contralateral (Fig 11A–11C). We quantified the number of slow fibers as a percentage of the total number of fibers in n = 5 animals per experimental group in the 6mo. recovery timepoint. We found that the mean percentage of the number of slow fibers was 37.1 ± 10.0% in the VML Only group and 40.0 ± 8.6% in the VML+SMU group, compared to the contralateral which was 27.9 ± 4.2%. However, these values were not significantly different (one-way ANOVA: P = 0.0807, n = 5 per group) (Fig 11D).

We also sought to determine the percentage of slow fibers constituting the total area of the muscle to give an indication as to the size of the fibers. The percentage of the area of slow fibers

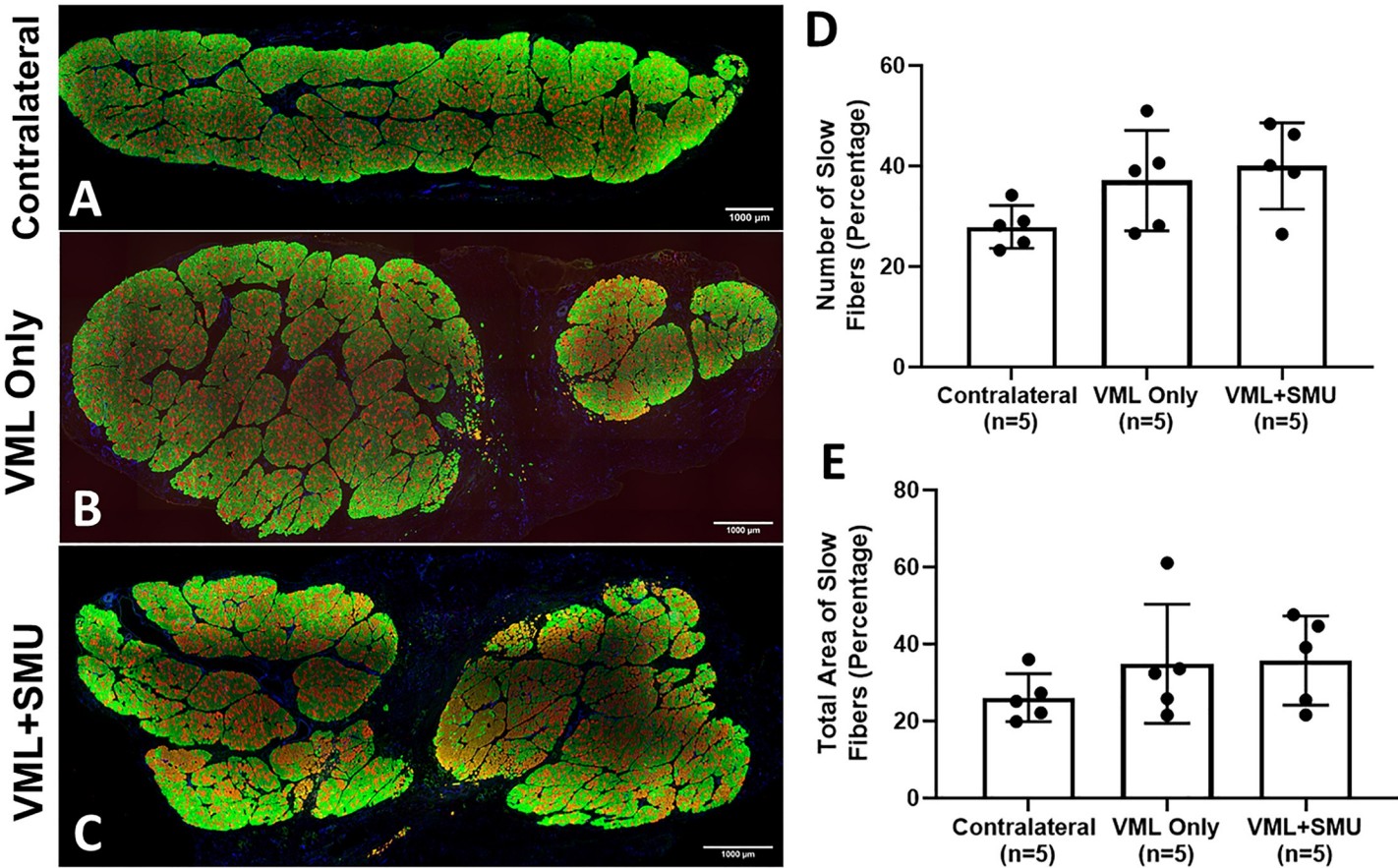

**Fig 11. Fiber type analysis at the 6-month recovery timepoint.** To note any changes to the fiber type of the msucle at the 6-month recovery timepoint, we performed immunostaining for fast myosin isoform (type II fibers, green) and slow myosin isoform (type I fibers, red) in the (A) contralateral, (B) VML Only, and (C) VML+SMU groups. Scale bars on A-C = 1000μm. (D) There was no significant difference in the percentage of the number of slow fibers between groups (P = 0.0807). (E) There was no significant difference in the percentage of the total area made up by slow fibers between groups (P = 0.3852).

was 34.9 ± 15.5% in the VML Only group and 35.7 ± 11.6% in the VML+SMU group, compared to the contralateral which was 26.1 ± 6.2%. Again, these values were not significantly different (P = 0.3852, n = 5 per group) (Fig 11E). We also noticed considerable slow (type I) fiber grouping (i.e. more than 50 grouped fibers present) in 40% of animals (n = 2/5) in the VML Only group and 100% of animals in the VML+SMU group (n = 5/5). Fiber grouping indicates that portions of the muscle that remained after VML were denervated at the time of injury and while they were reinnervated, likely contributed to the subsequent force deficits. A greater number of animals exhibited fiber type grouping in the VML+SMU group compared to the VML Only group which suggests that the injury induced denervation was more prevalent in this group, and thus the initial VML injury could have been more severe.

In the 6-month recovery group as a whole, the number of grouped fibers was not significantly correlated with the percentage of force recovery (r = -0.2488, P = 0.4883, n = 10) (Fig 12A). However, when looking at the experimental groups individually, the VML Only group experienced a significant negative correlation between the number of grouped fibers and the percentage of force recovery (r = -0.9506, P = 0.0131, n = 5) (Fig 12B). That is, an increased number of grouped fibers was correlated with a lower percentage of recovered force which suggests that the denervation injury negatively impacted the force recovery of the muscle. In contrast, there was a non-significant, much weaker correlation in the VML+SMU group (r =

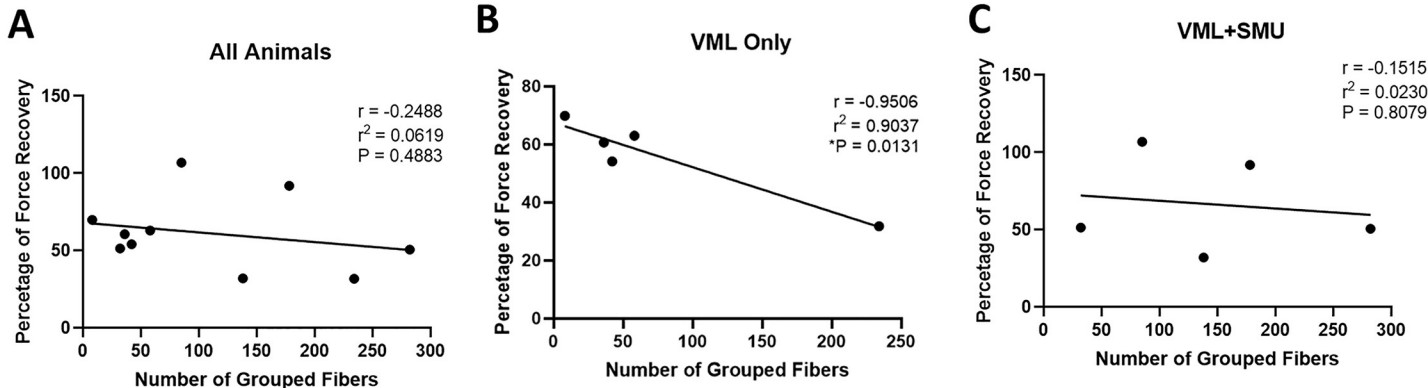

**Fig 12. Correlation between fiber grouping and force in the 6-month recovery group.** (A) There was no significant correlation between the number of grouped fibers and the percentage of force recovery in the 6-month recovery group as a whole (r = -0.2488, P = 0.4883, n = 10). (B) In the VML Only group, however, there was a significant negative correlation between the number of grouped fibers and the percentage of force recovery (r = -0.9506, P = 0.0131, n = 5). (C) In contrast, there was no significant correlation in the VML+SMU group (r = -0.1515, P = 0.8079, n = 5).

-0.1515, P = 0.8079, n = 5) (Fig 12C). This suggests that the presence of the SMU may have positively affected the force recovery of the muscle despite the evidence of extensive denervation, such that a high number of grouped fibers did not significantly correlate with low force recovery.

To evaluate the abundance of intramuscular fat, that is fat within the muscle tissue, we performed immunostaining for perilipin (Fig 13A–13I). A two-way ANOVA revealed that the recovery timepoint did not significantly affect the fat content of the muscles (P = 0.1396). Combining the recovery timepoints, differences in fat content between experimental groups were significant (one-way ANOVA: P = 0.0224), and the VML+SMU group had a significantly higher fat content than the contralateral muscles (TMC: P = 0.0334) (Fig 13J). Specifically, the fat content in the VML+SMU group was 2.07 ± 1.4% (n = 13), while the fat content was 1.09 ± 0.58% in the contralateral muscles (n = 12) and 1.18 ± 0.46% in the VML Only group (n = 12). Because increased intramuscular fat content is associated with various pathologies including denervation [41], the significantly higher fat content in the VML+SMU group corroborates the hypothesis that the denervation in the VML+SMU group was more severe than the denervation of the VML Only group.

## Discussion

This study aimed to evaluate the effects of our engineered skeletal muscle tissue in repairing a craniofacial VML injury. The literature suggests that craniofacial VML would manifest differently than trunk and extremity VML for a variety of reasons. This includes heterogeneity of both satellite cell populations [12,13] and ECM [10,11], differences in regenerative capacities [15], and differences in embryonic origin between craniofacial and trunk and limb muscle [16], as well as the documented complexity of craniofacial disorders relative to trunk and limb disorders [1]. Despite the notable differences between craniofacial muscle and trunk and limb muscle, all VML studies to date have involved VML models in trunk and extremity muscles [17]. This study sought to address this knowledge gap by introducing a large animal model of facial VML. Explicitly, the model used in this study was a 30% VML injury in the ovine zygomaticus major muscle.

Previous work in our lab compared the effects of using facial muscle-derived cells versus hindlimb muscle-derived cells in SMU fabrication [42]. Overall, we found that a hindlimb

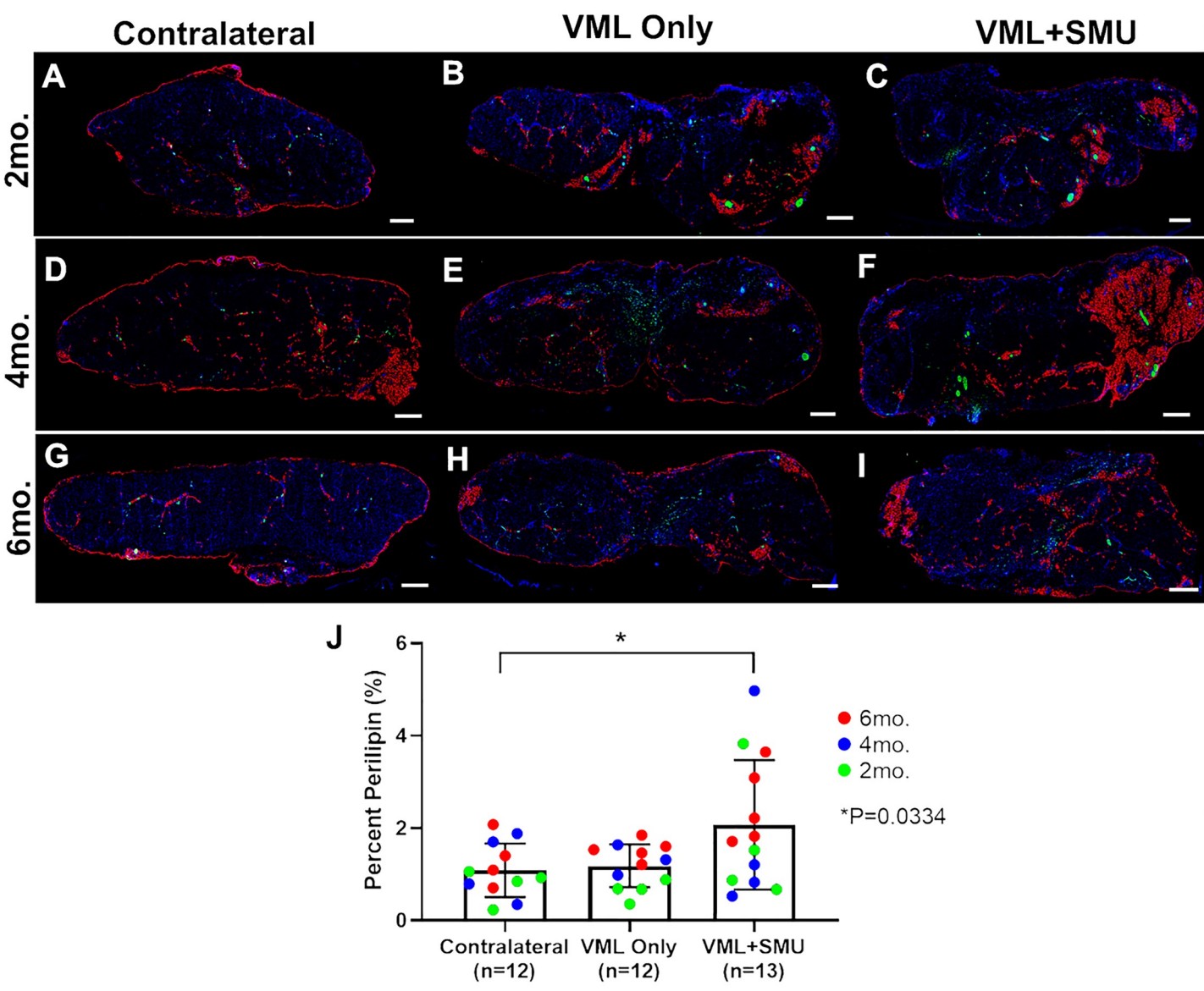

**Fig 13. Intramuscular fat.** (A-I) Immunostaining for neurofilament (SMI312, green) and fat (perilipin, red) in muscle cross-sections showed the presence of intramuscular fat and nerve in the explanted muscles. Scale bars = 1000μm. (J) There was a significantly higher amount of intramuscular fat in the VML+SMU group compared to the contralateral muscle (P = 0.0334), while there was no significant difference in the amount of intramuscular fat between the VML Only group and the contralateral (P = 0.9666).

muscle, the semimembranosus, was best-suited for SMU fabrication in that it was the most clinically relevant biopsy source and produced the most consistent SMUs with the highest myotube density [42]. This prompted the decision to use the semimembranosus muscle as the SMU cell source in this study. However, although these results informed our cell source, the combined effects of cell source and implantation site (i.e. facial muscle-derived SMUs implanted into hindlimb and facial muscle) is still unknown. Although one study found that masseter-derived satellite cells fused with irradiated limb muscle as efficiently as those derived from another limb muscle [13], no studies have compared regenerative capacities of facial and limb satellite cells when implanted into facial muscle. Facial muscle-derived cells can

contribute to limb regeneration, but craniofacial satellite cells have distinct gene expression profiles, respond differently to endogenous cues, and may operate under different molecular pathways [12]. In this study, hindlimb-derived SMUs were implanted into a facial VML injury; thus, there may have been a difference in regenerative outcomes if the SMUs had been fabricated from facial muscle-derived cells because the cells may have had greater regenerative potential in their source environment.

Although 30% of the muscle was removed, force capabilities ranged from 15% to 70% of the uninjured contralateral in the VML Only group compared to 28% to 107% in the VML+SMU group. The presence of force deficits exceeding the percentage of tissue removed was unsurprising, as force deficits greater than the loss of tissue are a hallmark of VML injuries and are evident both clinically and in animal models of VML [3,43]. Besides the loss of muscle tissue incurred by the initial injury, these force deficits can be exacerbated by damage to the remaining tissue. We observed a significant decrease in both the total ZM cross-sectional area and the MF20+ cross-sectional area in both experimental groups at all recovery timepoints. This was accompanied by a change in the composition of the ZM, with both experimental groups experiencing a significant decrease in the percentage of MF20+ tissue compared to the contralateral, as well as a significant decrease in the total muscle fiber counts. This loss of MF20+ tissue, as well as the increase in fibrotic connective tissue, contributed to the force deficits of the injured ZMs.

Additionally, muscle tethering and increased tissue stiffness are also noted contributors of the force deficits that accompany VML injuries and were qualitatively observed to varying degrees during biomechanical testing of the injured tissues. Changes in tissue quality were also evidenced by the significantly lower optimal length of the muscle fibers within the ZM in the VML+SMU group relative to the contralateral muscles. This was unsurprising, as decreased fiber length is suspected to occur following VML injury and decreasing the number of sarcomeres in series decreases the optimal length of the muscle as a whole [3,44].

The reported effects of denervation on muscle structure and function are numerous, and include muscle atrophy, sarcomeric disruption, and loss of specific force [41,45,46]. Evidence of denervation includes a shift in fiber type in that both fast and slow fibers to revert to a slow fiber type [47,48]. Indeed, we observed a non-significant increase in the abundance and total area made up by type I fibers in injured muscles relative to the contralateral. However, 70% of the animals had a quantity of type I fibers that exceeded the highest number exhibited by the contralateral. The shift towards a greater abundance of slow fibers could have resulted in reduced force capabilities by contributing to the reduction in MF20+ cross-sectional area, as type I fibers are typically smaller in diameter than type II fibers. Furthermore, type I fiber grouping is well-documented and almost universally accepted as a sign of denervation and subsequent reinnervation [49,50] and was observed in 70% of the animals in the 6-month recovery group. There was also as a significant negative correlation between the number of grouped fibers and the percentage of force recovery in the VML Only group such that a larger number of grouped fibers was significantly correlated with a lower percentage of force recovery. This suggests that the denervation and subsequent reinnervation evidenced by the notable fiber grouping likely contributed to the total force deficits experienced in the VML Only group.

There is also the possibility that the force deficits were exacerbated as the result of compensation of the contralateral ZM which would have resulted in its hypertrophy and increased the discrepancy in force production between the contralateral and injured ZMs. Although we did not observe any changes in motion that would have suggested compensation by the contralateral ZM, functional benchmarking prior to surgery and/or the use of sham animals will be taken into account in future studies.

While the SMU grafts did not result in significantly improved outcomes relative to the VML Only group, the SMUs did not appear to negatively impact the functional recovery. In fact, the mean force production and specific force (both as a percentage of the contralateral) was higher in the VML+SMU group compared to the VML Only group, although this difference was not significant. SMU efficacy may have been more apparent in a less severe injury model (i.e. without the comorbidities of ischemia and denervation). This is supported by the fiber grouping data, in which a significant negative correlation between the number of grouped fibers and the percentage of force recovery was observed in the VML Only group, but not in the VML+SMU group.

Our results also reaffirm that a nerve transfer with direct muscle neurotization is not sufficient to recover muscle structure and function following VML injury. A previous study in a rat VML model showed that even with a nerve re-route, the unrepaired VML group exhibited significantly lower maximum force production compared to the uninjured contralateral muscles [23]. Similarly, all animals in this study received a nerve re-route directly to the injury site, but this treatment was not enough to recover force production to the level of the contralateral. Furthermore, we did not see evidence of any fully formed neuromuscular junctions in the repair site at the 2mo. or 4mo. recovery timepoints, but we did note several NMJs in the repair site at the 6mo. recovery point in both experimental groups. However, even at 6 months, the presence of NMJs in the repair site was very limited, suggesting that a recovery period greater than 6-months is necessary to obtain functional innervation of the repair site in this facial VML model. Studies have shown that inflammatory environments can prevent nerve growth and result in the formation of neuromas if regeneration occurs during fibrotic tissue deposition [51]. Thus, inflammation may have prevented reinnervation of the injury site and contributed to the notable force deficits regardless of the length of the recovery period.

The geometry and location of the VML injury induced in this study may have created comorbidities that increased the severity of the injury and contributed to variability within groups. Specifically, as is common in extremity models, we chose to dissect a full thickness longitudinal portion of the ZM constituting 30% of the muscle mass (Fig 1A). In the native ZM, the vasculature inserts on the deep side of the muscle which made it difficult to create the VML injury without damaging the vascular bundle. Indeed, despite our attempts to avoid injuring native vasculature, in some instances, bleeding occurred during the muscle removal. We believe this injury led to ischemia that caused coagulative necrosis of the muscle in n = 5/30 animals, resulting in virtually no muscle fibers present in the midbelly of the ZM at the time of explant. It is well understood that ischemia leads to muscle necrosis, and total muscle necrosis can occur if ischemia occurs for prolonged periods of time [40,52]. Thus, the variability within groups may be related to variability in the severity of damage to the native vasculature. Moreover, if vascular damage led to necrosis of the remaining native muscle, it could have prevented survival of the SMU as well, as viability of implanted engineered tissues is dependent on rapid vascularization of the construct by the surrounding tissue [53].

Additionally, the results showed evidence of denervation and subsequent reinnervation. A study on the location of motor end plates (MEPs) in human facial muscle samples found that MEPs are located at four different places along the ZM (see Fig 2B of reference [39]). The location of our VML injury relative to the MEPs suggests that the muscle resection potentially created a denervation injury in the remaining muscle. This is corroborated by the notable type I fiber grouping, an indication of denervation and subsequent reinnervation [49,50], that was observed in n = 7/10 animals in the 6mo. recovery group. Because there are microscopic variations in the locations of native MEPs between individuals, the extent of the damage to the nerve was likely variable between animals which could explain the variability in the functional recovery of the animals within experimental groups.

Overall, our results suggest that variability in the severity of the initial injury may have differentially affected the experimental groups. Although there was no significant difference in the volume of muscle removed between experimental groups and between timepoints, it is possible that there were differences in the degree of denervation and/or ischemia both between and within groups. Specifically, the more prevalent type I fiber grouping in the 6mo. VML +SMU group and the significantly higher fat content of the VML+SMU group than the VML Only group implies that the acute denervation injury may have been more severe in the VML +SMU group. Variability in the severity of the injury could also explain why SMU mass did not correlate with functional outcomes and why the larger mass of the 6mo. SMUs did not result in improved functional recovery compared to the 2mo. and 4mo. groups.

Although the data showed high variability within experimental groups, inconsistency in the therapeutic response of VML treatments involving engineered muscle tissue is not unique to this study. A study by Corona et al. showed that only 46% of the animals treated with their engineered tissue exhibited significant functional recovery compared to the unrepaired controls [54]. They believe the reason for this variability was damage to the construct during handling and implantation [54]; however, this was addressed in a follow-up study and a variable response to VML repair was observed once again, despite improved outcomes (67% of animals demonstrated significant functional recovery) [55]. They suggested the reason for this was due to a change in the geometry of the VML injury model which used the same percentage of VML but changed the dimensions of the injury to make it more shallow [55]. They believe this geometry led to improved cell migration and nutrient flow; however, it is also possible that a shallower injury may have simply disrupted the native nerve and vasculature to a lesser extent than in their previous study. This highlights the difficulties associated with controlling the severity of a VML injury and with creating a model with a geometry that minimizes damage to native nerve and vasculature.

## Conclusions

In sum, while our injury model was representative of clinical manifestations of VML, the ischemia and denervation in addition to the loss of muscle was likely too severe of an injury for our engineered tissues to overcome. While the inclusion of comorbidities is more clinically realistic, they create additional variables to control for and can create a source of variation that makes it difficult to evaluate the efficacy of a therapy. For this reason, the majority of researchers choose not to include common comorbidities such as peripheral nerve injuries in their VML models [43]. In fact, a meta-analysis of the efficacy of various VML therapies chose not to consider the effects of denervation or vascular disruption in their analysis [17]. As a result, it is difficult to truly understand how engineered skeletal muscle tissue would perform in the treatment of clinical manifestations of craniofacial VML. This study highlights the importance of balancing the use of a clinically realistic model while also maintaining control over variables related to the severity of the injury. These variables include the volume of muscle removed, the location of the VML injury, and the geometry of the injury, as these affect both the muscle's ability to self-regenerate as well as the probability of success of the treatment. Future directions will move away from full-thickness VML models and utilize an injury geometry that avoids the bulk of native nerve and vasculature.

## Supporting information

**S1 Fig. Representative force tracings.** These images depict representative force tracings in response to a tetanic electrical stimulus in the contralateral ZM **(A,D,G),** the VML Only group **(B,E,H)**, and the VML+SMU group **(C,F,I)** as well as animals in the 2-month **(A-C),** 4-month

(**D-F**)**,** and 6-month (**G-I**) recovery timepoints.
(TIF)

## Acknowledgments

The authors would like to acknowledge Alexander Wood and Peter Macpherson for their contributions regarding technical expertise. Furthermore, the authors would like to acknowledge Charles Roehm (Engineering Design and Fabrication Core, University of Michigan Orthopedic Research Laboratory) as well as the Michigan Medicine Department of Pathology.

## Author Contributions

**Conceptualization:** Brittany L. Rodriguez, Paul S. Cederna, Lisa M. Larkin.

**Data curation:** Brittany L. Rodriguez, Emmanuel E. Vega-Soto, Christopher S. Kennedy, Matthew H. Nguyen.

**Formal analysis:** Brittany L. Rodriguez, Emmanuel E. Vega-Soto, Christopher S. Kennedy, Matthew H. Nguyen.

**Funding acquisition:** Lisa M. Larkin.

**Investigation:** Brittany L. Rodriguez, Emmanuel E. Vega-Soto, Paul S. Cederna, Lisa M. Larkin.

**Methodology:** Brittany L. Rodriguez, Paul S. Cederna, Lisa M. Larkin.

**Project administration:** Brittany L. Rodriguez.

**Writing – original draft:** Brittany L. Rodriguez.

**Writing – review & editing:** Brittany L. Rodriguez, Paul S. Cederna, Lisa M. Larkin.

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
