## [Decision Letter · Decision Letter 0]

29 May 2020

PONE-D-20-11394

A Tissue Engineering Approach for Repairing Craniofacial Volumetric Muscle Loss in a Sheep Following a 2, 4, and 6-Month Recovery

PLOS ONE

Dear Dr. Larkin,

Thank you for submitting your manuscript to PLOS ONE. After careful consideration, we feel that it has merit but does not fully meet PLOS ONE’s publication criteria as it currently stands. Therefore, we invite you to submit a revised version of the manuscript that addresses the points raised during the review process.

In particular, the reviewers have requested clarifications on some of the experimental details.  They have also raised concerns that some of the findings might be overstated and not completely supported by the data. The reviewers have also suggested further histologic analysis to help corroborate some of the conclusions.

We look forward to receiving your revised manuscript.

Kind regards,

Warren Grayson

Academic Editor

PLOS ONE

Journal Requirements:

Reviewers' comments:

Reviewer's Responses to Questions

**Comments to the Author**

1. Is the manuscript technically sound, and do the data support the conclusions?

Reviewer #1: Partly

Reviewer #2: Partly

2. Has the statistical analysis been performed appropriately and rigorously? 

Reviewer #1: Yes

Reviewer #2: Yes

3. Have the authors made all data underlying the findings in their manuscript fully available?

Reviewer #1: Yes

Reviewer #2: Yes

4. Is the manuscript presented in an intelligible fashion and written in standard English?

Reviewer #1: Yes

Reviewer #2: Yes

5. Review Comments to the Author

Reviewer #1: The manuscript by Rodriguez et al., evaluate a craniofacial VML injury and response to SMU repair that this research group has developed. First, the development of this new model is an important step in the field and should be commended. The terminal evaluations of this included (in comparison to the contralateral uninjured control), histologic and functional testing of the skeletal muscle. There are both minor and major issues that should be addressed.

Major:

1. The explanation related to embryologic origin of the limb vs. facial muscles is important to this work, and while noted in the introduction this is a topic that likely could also be addressed in the discussion. Additionally, the donor cell tissue (only female) was limb muscle is there is a justification for this and additionally do the authors think this affected the quality of the regeneration at all?

2. Related to the limb vs. facial muscle point above, have the authors considered evaluating the satellite cells of the muscle terminally? A good portion of the introduction comments on the proliferative capacity of limb vs. facial muscles and it would be useful to the analyses if even satellite cell histology were included in this work. With this, it would be useful to stain for both Pax7 and Pax3 in the muscles especially since cell populations (including satellite cells) from the limb muscle were used in the SMU construct in the current work. In the future, it may also be useful to evaluate the proliferative capacity of the satellite cells.

3. For the histologic procedures (line 242), when it notes muscles were divided into segments. Were there any full thickness sections of the midbelly saved? This would support the analysis of total muscle fiber number counting. I suspect there were since MyHC area was used to normalize force. It would be helpful for to evaluate the total number of midbelly muscle fibers. This can easily be evaluated using the count function in FIJI. As noted the ZM muscle not pennated and runs end-to-end to support this type of analysis.

4. Can the authors speculate the physiologic impact of the larger SMU at 6months?

5. Please identify which experimental groups had animals removed for clarity, line 401. This is of particular importance, since the removal of 5 animals is less than appear for the functional testing (total 23, not 25). Also please note how Fig 5 A, B have total of 23 animals, while C, D only have 22.

6. The addition of specific force in this work is useful. However, the MyHC positive area does not take into account the fibrosis which likely contributes to force generation. How has this been corrected for?

7. In Figure 9 G, this is the only panel in which the muscle is oriented longitudinally which is more acceptable to evaluated neuromuscular junctions, in this panel you can more clearly see the intramuscular nerve and at least one innervated end plate. Why were all the sections not evaluated in this orientation? It is difficult to agree with the interpretation with the images in this orientation, these are 10um sections for any of those the fibers (in red) could have a neuromuscular junction further down the length of those fibers, maybe just 10 or 20um away. Additionally, please identify the spatial location on the muscle the neuromuscular junctions were evaluated, as noted there are 4 identified areas (ref 39 in paper, line 688). Were these systematically evaluated in those areas, how many fibers or neuromuscular junctions were evaluated per group? This is a moderate measure of “innervation” and the authors should moderate the discussion of this.

8. Figure 10, the selected images are not representative of the data shown. For example, it is likely that the VML only image is the highest most data point in panel D. If possible, consider presenting images closer to the mean to be representative of the data. For D, can you clarify if this is total fiber number? With this, the area in panel E is really driven the distribution. It would be more informative to have mean cross-sectional fiber area, not total area. It is also possible that there are more slow fibers, then this is reporting by area since those fibers tend to be smaller than fast fibers.

9. For the evaluation of MyHC fiber typing, was any attempt to evaluated co-expressing fibers conducted? It is likely that fibers that undergo reinnervation by a differing motor neuron type are often co-expressing.

10. Fiber type grouping or clustering, is an interesting phenomena. There are many automated methods that can distinctly quantify this, many that interface with FIJI. Can the authors provide more clarity on how this was quantified and if it is qualitative herein? Often grouping does not need to be direct contact by just closer distance with a loss of the mosaic appearance of muscle.

11. The link between fiber type grouping and denervation is currently not well justified. Have the authors considered that slow fibers produce less force and the increase in percentage of slow fibers will affect overall force? (line 583-596)

12. Figure 12, the histology shows that the fascia picks up some of the fat staining. Was this excluded in the analysis? Additionally, were sections evaluated closer to support “intramuscular” or fat in each muscle fiber?

13. The discussion could benefit from decreasing the link between fiber type grouping, innervation, and functional outcomes. The data in support of this here in are not as robust as the discussion presents them. One spot for example, line 665 “fully formed neuromuscular junctions…” this was really not evaluated, they were just moderately evaluated as present. The authors should revaluate statements throughout the discussion and make sure they are supported by the work.

Minor:

1. There were references that did not format correctly, lines 421 and 606.

2. Please note the sex of the experimental animals, line 190.

3. Please add the absolute weight of the removed muscle tissue, line 338.

4. In line 48, please consider removing “degenerative” as those types of disorders often have a different etiology than traumatic VML.

5. Since this is a new model, it would be useful to the reader to include force tracings of all experimental groups.

6. What was the cut off to consider a muscle fiber “small”, line 263 and 523? Was this uniformly set in FIJI? In other large animals studies less than 500um2 is often used.

7. Figure 2, please clarify what the red data points are in panel C.

8. Figure 4, it could help readability if the time points were labeled near the A,B,C on the figure.

9. Line 477, please clarify is this muscle fiber cross sectional area? Or area of MF20 positive fibers?

10. Line 533, was there any attempt to quantify how many centrally located nuclei there were?

Reviewer #2: In the article, titled “A tissue engineering approach for repairing craniofacial volumetric muscle loss in a sheep following a 2, 4, and 6-month recovery”, the authors generate one of the first craniofacial VML defect models in sheep and investigate the ability of skeletal muscle units (SMUs) to enhance recovery. The craniofacial VML model was demonstrated to generate complex injuries including vascular and neural damage, which confounded results investigating the contribution of SMUs to the repairing muscle. Overall, the study is of significant interest to the skeletal muscle community and speaks to issues regarding injury scaling and the treatment of craniofacial muscles. However, there are some concerns with data presentation and analysis that undermine the conclusions presented by the authors.

Methods:

1. Do the authors have data from age-matched control animals? In using the contralateral muscle as a control, there are potential issues with muscle hypertrophy or compensation via overuse of the contralateral. This could further obfuscate the contribution of SMUs to muscle repair. Commenting on this in the discussion may also be appropriate.

2. How many slides/sections per animal were used to obtain IHC results (must use multiple sections per analysis)?

Results:

3. Page 16, line 332: Mere presence of DAPI-stained nuclei does not eliminate the possibility of a necrotic core. The authors should present higher magnification images of the central regions of SMUs to assess nuclear morphology to claim lack of cell necrosis.

4. Page 17, lines 345-48: What was the rationale for implanting larger constructs into a single treatment group, as opposed to spreading them out across time points? Did “sentinel” SMUs from this group show differences in morphology? Without knowing more about these groups, it is unclear whether VML+SMU groups rescued patients/muscles from otherwise more severe injuries or not.

5. Along the same line as above, did the larger SMUs have increased force production (or specific force)?

6. Page 20, line 421: Please correct/remove this error message, and supply the intended text.

7. Page 29, line 606: Please correct/remove this error message, and supply the intended text.

8. Did the authors see histologic evidence of the SMUs at any time point? The authors’ previous work has shown SMU integration with the host, but usually they are distinguishable from host muscle. If the SMUs are not present, do the authors have a hypothesis as to why?

9. In conjunction with the analysis of fat content in the muscles (page 29, lines 605-616), did the amount of collagen (i.e. scar formation) change between treatment groups (i.e. quantification of Fig 7D-F)?

10. Did the authors note any differences in muscle repair/phenotype as a function of depth within the tissue?

Discussion:

11. Page 30, line 628: Sentence ends with hanging clause “…reasons.” Please finish the sentence, copying from the introduction if necessary.

12. Page 31, line 641: It is not wholly clear why the benchmark of 70% was selected to be a comparison of note. Likely, the rationale is that this percentage would indicate a simple “increase” in the force production beyond a theoretical loss of 30% volume. However, the authors, as well as the field, acknowledge that this assumption of a linear loss of function with respect to defect size is incorrect and not valid. The authors should omit this comparison, or focus on the patients with ~100% force production with respect to the contralateral.

13. Page 32, lines 668-671: The authors make an interesting point regarding the role of inflammation in the repair of VML defects. Did the authors assess (qualitatively or quantitatively) inflammation within these sites? An assessment of chronic inflammation may strengthen these claims.

14. Page 32, line 672: Reference 24 removed 5.26 +/- 0.76 g of the peroneus tertius muscle, whereas this study removed ~ 0.6-0.72 g (exact mass not reported) of the zygomaticus major muscle. These are not “identical”. While the intent of the authors might be that the percentage of tissue that was removed was similar (~30%), the volume is an order of magnitude different. While this may further support the authors narrative (i.e. craniofacial VML is more complex), the authors should revise this paragraph to more concisely make their comparisons and derive their conclusions.

15. The authors posit that ischemic conditions may also be affecting the functional recovery of the injuries. Did they quantify the amount of blood vessels in the wound sites and if so, were differences observed?

Figures

16. Fig 1D – Did the authors stitch the remaining muscle together with the SMUs after the generation of the VML – is that what this image is showing? If so, was this similarly done (or why not done) in the VML only condition?

17. Fig 2B – Please increase contrast/visibility of myofibers in this image.

18. Fig 2F/G – The text states that n=2 SMUs were used in implantation studies, but there appear to be 3 discrete “bundles” in these images. Are there 3 SMUs?

19. Fig 9 – How can you parse the expression of synaptic vesicle protein-2 against neurofilament, when they are visualized with the same fluorophore? Are some images visualizing one marker over the other? In general, the quality of these images could be improved. Interpretation of what are NMJs is unclear, the authors could indicate representative NMJs in the images, as this becomes important within the discussion (page 32, lines 665-668)

20. Fig 12 – Please increase the contrast of images. While the reviewer appreciates the low percentage of fat tissue present in the samples, the quality of these images could be improved.

6. PLOS authors have the option to publish the peer review history of their article (what does this mean?). If published, this will include your full peer review and any attached files.

Reviewer #1: No

Reviewer #2: No

---

## [Author Response · Author response to Decision Letter 0]

19 Jul 2020

Dear Reviewers,

First, thank you for taking the time to review our work. Your feedback was thorough and greatly appreciated. All changes to the manuscript have been highlighted in yellow.

Comments to the Author

Reviewer #1: The manuscript by Rodriguez et al., evaluate a craniofacial VML injury and response to SMU repair that this research group has developed. First, the development of this new model is an important step in the field and should be commended. The terminal evaluations of this included (in comparison to the contralateral uninjured control), histologic and functional testing of the skeletal muscle. There are both minor and major issues that should be addressed.

Major:

Comment 1. The explanation related to embryologic origin of the limb vs. facial muscles is important to this work, and while noted in the introduction this is a topic that likely could also be addressed in the discussion. Additionally, the donor cell tissue (only female) was limb muscle is there is a justification for this and additionally do the authors think this affected the quality of the regeneration at all?

Response: The reason we had chosen to implant female cells into male animals was so we could perform cell tracking. Unfortunately, given technical difficulties, we were not able to perform cell tracking (through genotyping) in this study, but we hope to be able to track the integration of the SMU into the host in future large animal studies. Although the sex of the donor and recipient animals differed, we did not observe evidence of an adaptive immune response, but it is possible that the use of female-derived SMUs contributed to an inflammatory response that was not specifically measured in the bloodwork. This was clarified in the results, line 416: “Notably, although the SMUs were allografts derived from female donors, none of the animals exhibited signs of a chronic immune response at the time of explant (i.e. white blood cell counts were within normal limits). However, the use of allografts may have contributed to an inflammatory response that was not specifically measured in the bloodwork.”

With regard to the use of hindlimb-derived cells, previous work in our lab studied the effects of muscle cell source on the myogenesis, structure and function of our SMUs. In a previous study (Rodriguez, BL et al, A Comparison of Ovine Facial and Limb Muscle as a Primary Cell Source for Engineered Skeletal Muscle. Tissue Eng Part A. (2019)), we found that the semimembranosus muscle was best suited as a cell source for SMU fabrication, whereas the facial muscles studied had significant disadvantages. Thus, from the perspective of clinical relevance and in vitro measures of success, we chose to use hindlimb-derived SMUs for our large animal implantation studies. The following paragraph was added to the discussion to explain the choice of hindlimb-derived SMUs and the possible effects of implanting hindlimb-derived cells into facial muscle: 

“Previous work in our lab compared the effects of using facial muscle-derived cells versus hindlimb muscle-derived cells in SMU fabrication [42]. Overall, we found that a hindlimb muscle, the semimembranosus, was best-suited for SMU fabrication in that it was the most clinically relevant biopsy source and produced the most consistent SMUs with the highest myotube density [42]. This prompted the decision to use the semimembranosus muscle as the SMU cell source in this study. However, although these results informed our cell source, the combined effects of cell source and implantation site (i.e. facial muscle-derived SMUs implanted into hindlimb and facial muscle) is still unknown. Although one study found that masseter-derived satellite cells fused with irradiated limb muscle as efficiently as those derived from another limb muscle [13], no studies have compared regenerative capacities of facial and limb satellite cells when implanted into facial muscle. Facial muscle-derived cells can contribute to limb regeneration, but craniofacial satellite cells have distinct gene expression profiles, respond differently to endogenous cues, and may operate under different molecular pathways [12]. In this study, hindlimb-derived SMUs were implanted into a facial VML injury; thus, there may have been a difference in regenerative outcomes if the SMUs had been fabricated from facial muscle-derived cells because the cells may have had greater regenerative potential in their source environment.” (lines 755-771)

Comment 2. Related to the limb vs. facial muscle point above, have the authors considered evaluating the satellite cells of the muscle terminally? A good portion of the introduction comments on the proliferative capacity of limb vs. facial muscles and it would be useful to the analyses if even satellite cell histology were included in this work. With this, it would be useful to stain for both Pax7 and Pax3 in the muscles especially since cell populations (including satellite cells) from the limb muscle were used in the SMU construct in the current work. In the future, it may also be useful to evaluate the proliferative capacity of the satellite cells. 

Response: We agree, satellite cell histology would have been a valuable addition to the implantation studies and would also give insight into the regeneration of the tissue. We will take this into account in future studies.

Comment 3. For the histologic procedures (line 242), when it notes muscles were divided into segments. Were there any full thickness sections of the midbelly saved? This would support the analysis of total muscle fiber number counting. I suspect there were since MyHC area was used to normalize force. It would be helpful for to evaluate the total number of midbelly muscle fibers. This can easily be evaluated using the count function in FIJI. As noted the ZM muscle not pennated and runs end-to-end to support this type of analysis.

Response: We have added in the data regarding total muscle fiber counts.

The following was added to the methods, line 270-272: “Additionally, total fiber counts were taken for each midbelly ZM cross-section. Using the MF20 and laminin-stained midbelly cross-sections sections, the total fiber count for each muscle was enumerated using ImageJ/Fiji.”

The following was added to the results: We also took a measure of total fiber counts in cross-sections of the ZM midbelly. A two-way ANOVA revealed that there was no significant difference in the total fiber counts as a percentage of the contralateral ZM between recovery timepoints (P=0.4046) or between experimental groups (P=0.7197). Specifically, the mean fiber count as a percentage of the contralateral was 55.7 ± 19.3% in the VML Only group and 59.4 ± 13.7% in the VML+SMU group. Given that the timepoints did not significantly affect normalized specific force, we combined all timepoints and found there was no significant difference in the total number of muscle fibers between experimental groups (P=0.4054) but the total number of muscle fibers was significantly lower in the surgical ZMs compared to the contralateral (P<0.0001) in both experimental groups. This indicates that the loss of MF20+ cross-sectional area in the injured ZMs was consistent with a loss in muscle fiber number. (lines 548-558).

Fig. 6. MF20+ Area, Specific Force, and Fiber Counts . (E) A two-way ANOVA showed there was no significant difference in normalized muscle fiber counts between recovery timepoints (P=0.4046) or between experimental groups (P=0.7197). (F) In both experimental groups, the total number of muscle fibers in the surgical ZM was significantly lower than the uninjured contralateral (P<0.0001 in both groups, n=11 for VML Only, n=12 for VML+SMU). 

Comment 4. Can the authors speculate the physiologic impact of the larger SMU at 6months?

Response: Despite being larger, there were no apparent differences in the monolayer quality, morphology, or force production of the SMUs implanted into the 6-month recovery group animals. Because there is no correlation between SMU size and force recovery (discussed in lines 867-869), it is difficult to determine the overall effect of this extra SMU mass. 

The additional mass of these SMUs was due to the growth of an additional monolayer on the plate that subsequently delaminated and contributed extra mass to the SMU. This monolayer would have a similar composition to the other monolayers. Specifically, it would be composed of myotubes, fibroblasts and other connective tissue cells, and ECM protein.

The additional ECM protein and connective tissue-secreting cells would have contributed scaffold material to the defect onto which native cells could grow. Conversely, the additional connective tissue cells could have negatively contributed to the fibrotic response. It is possible that these cells received pro-inflammatory signals and secreted additional proteins that contributed to fibrosis.

It is also likely that the larger SMUs contributed more myotubes to the repair. These cells, if they survived implantation, would have donated new muscle fibers to the repair site, which would offset some of the functional deficit created by the injury.

The following was added to the manuscript, line 398-401: “It is important to note that although larger, these SMUs did not exhibit any differences in monolayer quality, morphology, or force production compared to the other sentinel SMUs. Thus, these SMUs likely contributed additional ECM protein, connective tissue cells, and myotubes to the repair site.”

Comment 5. Please identify which experimental groups had animals removed for clarity, line 401. This is of particular importance, since the removal of 5 animals is less than appear for the functional testing (total 23, not 25). Also please note how Fig 5 A, B have total of 23 animals, while C, D only have 22.

Response: The following was added to the results to clarify which animals were excluded from the histological analysis: “Specifically, out of the five animals exhibiting coagulative necrosis, n=3 were part of the 4-month recovery timepoint (n=2 VML Only, n=1 VML+SMU) and n=2 animals were part of the 2-month recovery timepoint (n=1 VML Only, n=1 VML+SMU).” (line 438-440).

An additional n=2 animals (total of n=23) were excluded from functional (biomechanical) evaluations because we failed to get accurate force measurements. This was clarified in the manuscript: “An additional n=2 animals were excluded from biomechanical analyses (n=1 VML Only, 4-month timepoint and n=1 VML+SMU, 2-month timepoint) because we failed to collect accurate force data for those animals; however, the explanted muscles were still used in histological analyses.” (lines 441-444).

There is n=1 animal fewer in the optimal length measurements (Fig 5C,D) (total of n=22) because we failed to record the optimal length of the muscle for that animal. The muscle had been explanted by the time we realized we had not recorded the optimal length.

Comment 6. The addition of specific force in this work is useful. However, the MyHC positive area does not take into account the fibrosis which likely contributes to force generation. How has this been corrected for?

Response: To account for fibrosis, we have measured the total cross-sectional area of the ZM and subtracted the MF20+ area to calculate the remaining area (primarily the fibrotic connective tissue). The total cross-sectional area was used to calculate an additional specific force value and has been described thoroughly in the manuscript.

The following has been added to the methods, lines 273-286: 

“To quantify the total cross-sectional area of the muscle, including both MF20+ muscle fibers as well as connective tissue, nerve, and blood vessels within the muscle, the same MF20 and laminin-stained midbelly ZM cross-sections were used. The total cross-sectional area was measured in ImageJ/Fiji using the freehand selection tool. This total cross-sectional area was then used to calculate the total cross-sectional area specific force (N/cm2) by dividing the maximum tetanic force of the muscle by its total cross-sectional area. The comparison of the MF20+ specific force and the total cross-sectional area specific force gives insight into the change in muscle content, including the increase in fibrotic connective tissue in the injured muscles. 

 Finally, to quantify the difference in the amount of connective tissue between the uninjured contralateral muscles and the injured ZMs, we calculated the MF20- area by subtracting the MF20+ area from the total cross-sectional area and represented it as a percentage of the total cross-sectional area. This portion of the total area that was made up of connective tissue was then normalized to the amount that was present in the contralateral muscle.”

This has been added to the results, lines 559-616: 

“We calculated the total cross-sectional area of the injured and uninjured ZMs to evaluate whether there were changes in the muscle size as a result of the VML injury. We normalized the total cross-sectional area of the surgical ZM to the contralateral and found that there was no significant difference in normalized MF20+ area between recovery timepoints (P=0.2258) or between experimental groups (P=0.6753). The average normalized total cross-sectional area was 74.0 ± 27.2% in the VML Only group (n=11) and 79.6 ± 15.1% in the VML+SMU group (n=12) (Fig 7A). Combining all timepoints, there was no significant difference in total cross-sectional area between the experimental groups (two-way RM ANOVA: P=0.5769), but there was a significant difference between the contralateral and surgical ZMs (two-way RM ANOVA: P<0.0001) (Fig 7B). Using a Sidak’s multiple comparisons test, there was a significant difference between the total cross-sectional area in the contralateral and surgical ZMs of both experimental groups (P=0.0008, n=11 for VML Only; P=0.0070, n=12 for VML+SMU) indicating loss of ZM cross-sectional area in both experimental groups at the time of explant with respect to the uninjured contralateral.

The specific force calculated by using the MF20+ cross-sectional area revealed that there was no significant difference between the injured and uninjured ZMs which indicates that the remaining muscle tissue was healthy. However, calculating specific force using the total cross-sectional area of the muscle, which includes the muscle fibers as well as connective tissue, reveals the overall tissue quality of the ZM. Using the total cross-sectional area specific force and normalizing it to the contralateral, the mean normalized total cross-sectional area specific force was 74.8 ± 32.5% in the VML Only group (n=11) and 77.5 ± 27.6% in the VML+SMU group (n=12) (Fig 7C). A two-way ANOVA showed that there was no significant difference in the total-cross sectional area specific force as a percentage of the contralateral between experimental groups (P=0.6753). Combining all timepoints, there was no significant difference in the total cross-sectional area specific force between experimental groups (P=0.2594) but there was a difference between the contralateral and surgical ZMs (P=0.0009) using a paired two-way ANOVA. Notably, there was a significant difference in the specific force between the surgical ZM and the uninjured contralateral ZM in both the VML Only (P=0.0384, n=11) and the VML+SMU (P=0.0165, n=12) group (Fig 7D). This indicates that the functional capacity of the ZM as a whole was decreased as a result of the VML injury. 

To quantify the amount of connective tissue present in the ZM, that is MF20- tissue, we subtracted the MF20+ area from the total cross-sectional area and represented this as a percentage of the total area. When normalizing the percentage of MF20- tissue to the control, a two-way ANOVA revealed that there was no significant difference between recovery timepoints (P=0.1138) or between experimental groups (P=0.1016). Specifically, the MF20- area was 156 ± 48.4% of the contralateral in the VML Only group and 187 ± 53.7% of the contralateral in the VML+SMU group, indicating an over 50% increase in connective tissue in the injured ZMs (Fig 7E). Combining all recovery timepoints, there was no significant difference between experimental groups (P=0.6133) but there was a significant difference between the contralateral and surgical ZMs (P<0.0001). Specifically, the percentage of the cross-sectional area that was MF20- was 26.3 ± 5.5%, 41.7 ± 12.9%, 46.3 ± 12.0% in the contralateral, VML Only, and VML+SMU groups, respectively (Fig 7F). Overall, this indicates a significant increase in the connective tissue content of the injured ZMs of both experimental groups at all recovery timepoints.”

The additional figure and figure legend, for your reference:

Fig 7. Total Cross-Sectional Area and Specific Force. (A) A two-way ANOVA revealed that there was no significant difference in the total cross-sectional area as a percentage of the contralateral between experimental groups (P=0.6753). (B) In both experimental groups, the total cross-sectional area of the surgical ZM was significantly lower than the uninjured contralateral (P=0.0008, n=11 for VML Only; P=0.0070, n=12 for VML+SMU). (C) There was no significant difference in normalized total cross-sectional area specific force between experimental groups (P=0.0194). (D) In both experimental groups, the total cross-sectional area specific force was significantly lower in the surgical ZM than the uninjured contralateral (P=0.0384, n=11 for VML Only; P=0.0165, n=12 for VML+SMU). (E) There was no significant difference in the MF20- area as a percentage of the contralateral between recovery timepoints (P=0.1138) or between experimental groups (P=0.101). (F) In both experimental groups, the percentage of the total cross-sectional area that did not stain for MF20 was significantly higher than the uninjured contralateral (P=0.0007, n=11 for VML Only; P<0.0001, n=12 for VML+SMU).

Comment 7. In Figure 9 G, this is the only panel in which the muscle is oriented longitudinally which is more acceptable to evaluated neuromuscular junctions, in this panel you can more clearly see the intramuscular nerve and at least one innervated end plate. Why were all the sections not evaluated in this orientation? It is difficult to agree with the interpretation with the images in this orientation, these are 10um sections for any of those the fibers (in red) could have a neuromuscular junction further down the length of those fibers, maybe just 10 or 20um away. Additionally, please identify the spatial location on the muscle the neuromuscular junctions were evaluated, as noted there are 4 identified areas (ref 39 in paper, line 688). Were these systematically evaluated in those areas, how many fibers or neuromuscular junctions were evaluated per group? This is a moderate measure of “innervation” and the authors should moderate the discussion of this.

Response: While it appears that some of the images are not oriented longitudinally, this is due to the disorganized muscle fibers present in the repair site. All sections used in this figure were sectioned along the length of the muscle fibers. The repair site was often characterized by the presence of muscle fibers that were not oriented in parallel with the rest of the muscle, which is why some of the fibers appear in cross-section. Additionally, the entire width of the repair site and about cm of its length were evaluated, not just the 4 regions used in other measures. These sections are 25um; thank you for pointing out that this had not been stated in the methods. It has been clarified: “Longitudinal samples comprising the entire width of the ZM and ~2cm of the length were sectioned at 25μm and immunohistochemically stained…” (line 254-258). 

We agree, this method of identifying NMJs is purely qualitative and meant to provide visual support of the denervation hypothesis. We have clarified the results and discussion to better explain our interpretation of the NMJ histology. 

In the results: “We also immunohistochemically stained for the presence of neuromuscular junctions (NMJs) in longitudinal sections of the injury site and in the uninjured contralateral muscles (Fig 9). An NMJ was noted to be present if there was an overlap of the pre-synaptic and post-synaptic structures which resulted in a yellow region due to the overlap of the red and green fluorophores. This overlapping of structures was observed in the contralateral ZM at all recovery timepoints, and at least one NMJ was identified in the repair site of both the VML Only and the VML+SMU groups at the 6mo. timepoint. Although acetyl choline receptors and neurofilament were present at earlier timepoints, small muscle fibers in the injury site do not appear to be reinnervated (no overlapping structures) in the 2mo. and 4mo. recovery groups.” (lines 658-666)

In the discussion: “Furthermore, we did not see evidence of any fully formed neuromuscular junctions in the repair site at the 2mo. or 4mo. recovery timepoints, but we did note several NMJs in the repair site at the 6mo. recovery point in both experimental groups. However, even at 6 months, the presence of NMJs in the repair site was limited, suggesting that a recovery period greater than 6-months is necessary to obtain functional reinnervation in this facial VML model.” (lines 826-830)

Comment 8. Figure 10, the selected images are not representative of the data shown. For example, it is likely that the VML only image is the highest most data point in panel D. If possible, consider presenting images closer to the mean to be representative of the data. For D, can you clarify if this is total fiber number? With this, the area in panel E is really driven the distribution. It would be more informative to have mean cross-sectional fiber area, not total area. It is also possible that there are more slow fibers, then this is reporting by area since those fibers tend to be smaller than fast fibers.

Response: The figure has been updated to depict images that are more representative of the data. It is shown below for your reference.

For panel D, “four regions that were 9.6mm2 in area were chosen at random. In total, these areas constituted ~30% of the total muscle cross-sectional area. The number of fibers expressing slow myosin isoform (type I fibers) and the number of fibers expressing fast myosin isoform (type II fibers) were enumerated. This data was presented as the total number of slow fibers as a percentage of the total number of fibers enumerated.” 

In contrast, for panel E, the entire muscle cross-section was evaluated. We understand that slow fibers are usually smaller which is why we chose to present both fiber counts and area consisting of slow fibers. This is reflected by the fact that although there was an increase in total area of slow fibers with respect to the contralateral, there was an even greater increase in the number of slow fibers with respect to the contralateral. This implies that the slow fibers are smaller.

Comment 9. For the evaluation of MyHC fiber typing, was any attempt to evaluated co-expressing fibers conducted? It is likely that fibers that undergo reinnervation by a differing motor neuron type are often co-expressing.

Response: We did note the presence of co-expressing fibers in the muscles; however this number was typically less than 1 hybrid fiber per mm2. These fibers were noted in both the injured muscles and the contralaterals, but this was unsurprising as the presence of hybrid fibers is actually a unique characteristic of facial muscle [1-3].

1. Garland, C. and J. Pomerantz, Regenerative Strategies for Craniofacial Disorders. Frontiers in Physiology, 2012. 3(453).

2. Stål, P., Characterization of human oro-facial and masticatory muscles with respect to fibre types, myosins and capillaries. Morphological, enzyme-histochemical, immuno-histochemical and biochemical investigations. Swed Dent J Suppl, 1994. 98: p. 1-55.

3. Porter, J.D., et al., Distinctive morphological and gene/protein expression signatures during myogenesis in novel cell lines from extraocular and hindlimb muscle. Physiol Genomics, 2006. 24(3): p. 264-75.

Comment 10. Fiber type grouping, or clustering, is an interesting phenomena. There are many automated methods that can distinctly quantify this, many that interface with FIJI. Can the authors provide more clarity on how this was quantified and if it is qualitative herein? Often grouping does not need to be direct contact by just closer distance with a loss of the mosaic appearance of muscle.

Response: For this evaluation, we defined a grouped fiber as a type I muscle fiber that was completely surrounded by other type I muscle fibers. We then enumerated the number of grouped fibers present in the entire muscle cross section. This was clarified in the methods, lines 298-300: “To evaluate the degree of fiber type grouping, we also enumerated the number of grouped fibers in the entire muscle cross-section. A “grouped fiber” is defined as a type I muscle fiber that is completely surrounded by other type I muscle fibers.”

We acknowledge that this definition of a grouped fiber, compared to more loose definitions, underestimates the fiber grouping present in these muscles. However, this was the most quantitative method available to us. 

Comment 11. The link between fiber type grouping and denervation is currently not well justified. Have the authors considered that slow fibers produce less force and the increase in percentage of slow fibers will affect overall force? (line 583-596)

Response: It is well-reported that denervation causes both fast and slow fibers to revert to a slow fiber type [4, 5]. Because there is an increase in both the percentage by number and the percentage by area of slow fibers with respect to the contralateral, this presents evidence of denervation and reinnervation by a slow motor unit. Furthermore, type I fiber grouping is well-documented and almost universally accepted as a sign of denervation and subsequent reinnervation [6, 7]. 

Slow fibers do produce less force, but this is due to their lower cross-sectional area compared to fast fibers. The specific force is not different between fiber types [8]. Thus, the reduced total muscle CSA could be in part due to fiber type switching which would reduce total force by reducing the total muscle cross-sectional area. Denervation also causes a decrease in fiber cross-sectional area (fiber atrophy). In terms of the reduction in force with respect to the contralateral muscle, there are likely many contributing factors: loss of muscle tissue due to VML injury, muscle tethering and fibrosis inhibiting motion, atrophy of the fibers due to denervation/peripheral nerve damage, and denervation-related loss of specific force [9, 10].

These points were added to the discussion; please see the response to comment 13 below. 

4. d'Albis, A., et al., The effect of denervation on myosin isoform synthesis in rabbit slow-type and fast-type muscles during terminal differentiation. Denervation induces differentiation into slow-type muscles. Eur J Biochem, 1994. 223(1): p. 249-58.

5. Leeuw, T., M. Kapp, and D. Pette, Role of innervation for development and maintenance of troponin subunit isoform patterns in fast- and slow-twitch muscles of the rabbit. Differentiation, 1994. 55(3): p. 193-201.

6. Campbell, W.W., Evaluation and management of peripheral nerve injury. Clin Neurophysiol, 2008. 119(9): p. 1951-65.

7. Gordon, T. and J.E.T. de Zepetnek, Motor unit and muscle fiber type grouping after peripheral nerve injury in the rat. Exp Neurol, 2016. 285(Pt A): p. 24-40.

8. Lucas, S.M., R.L. Ruff, and M.D. Binder, Specific tension measurements in single soleus and medial gastrocnemius muscle fibers of the cat. Exp Neurol, 1987. 95(1): p. 142-54.

9. Geiger, P.C., et al., Effect of unilateral denervation on maximum specific force in rat diaphragm muscle fibers. J Appl Physiol (1985), 2001. 90(4): p. 1196-204.

10. van der Meulen, J.H., et al., Denervated muscle fibers explain the deficit in specific force following reinnervation of the rat extensor digitorum longus muscle. Plast Reconstr Surg, 2003. 112(5): p. 1336-46.

Comment 12. Figure 12, the histology shows that the fascia picks up some of the fat staining. Was this excluded in the analysis? Additionally, were sections evaluated closer to support “intramuscular” or fat in each muscle fiber?

Response: Yes, this false-positive staining was excluded from the analysis. This was clarified in the methods, lines 306-307. “False-positive staining of the muscle fascia as well as “edge effect” staining was excluded from this analysis.” 

We apologize for the confusion regarding the term “intramuscular fat”. By intramuscular, we mean fat within the muscle tissue, rather than fat within the muscle fibers. We did not evaluate whether there was an increase in the intracellular fat, but we will consider this in future experiments. This was clarified in the manuscript, line 724-725. “To evaluate the abundance of intramuscular fat, that is fat within the muscle tissue, we performed immunostaining for perilipin (Fig 12A-I).”

Comment 13. The discussion could benefit from decreasing the link between fiber type grouping, innervation, and functional outcomes. The data in support of this here in are not as robust as the discussion presents them. One spot for example, line 665 “fully formed neuromuscular junctions…” this was really not evaluated, they were just moderately evaluated as present. The authors should revaluate statements throughout the discussion and make sure they are supported by the work.

Response: We agree, the portion of the discussion regarding NMJs was explained poorly initially. We meant to highlight the lack of NMJs, rather than their presence. This has been revised. Please see response to comment 7.

The link between functional outcomes, fiber grouping, and fiber type shift was explained more clearly in the discussion, lines 772-804:

“Although 30% of the muscle was removed, force capabilities ranged from 15% to 70% of the uninjured contralateral in the VML Only group compared to 28% to 107% in the VML+SMU group. The presence of force deficits exceeding the percentage of tissue removed was unsurprising, as force deficits greater than the loss of tissue are a hallmark of VML injuries and are evident both clinically and in animal models of VML [3, 43]. Besides the loss of muscle tissue incurred by the initial injury, these force deficits can be exacerbated by damage to the remaining tissue. We observed a significant decrease in both the total ZM cross-sectional area and the MF20+ cross-sectional area in both experimental groups at all recovery timepoints. This was accompanied by a change in the composition of the ZM, with both experimental groups experiencing a significant decrease in the percentage of MF20+ tissue compared to the contralateral, as well as a significant decrease in the total muscle fiber counts. This loss of MF20+ tissue, as well as the increase in fibrotic connective tissue, contributed to the force deficits of the injured ZMs. 

Additionally, muscle tethering and increased tissue stiffness are also noted contributors of the force deficits that accompany VML injuries and were qualitatively observed to varying degrees during biomechanical testing of the injured tissues. Changes in tissue quality were also evidenced by the significantly lower optimal length of the muscle fibers within the ZM in the VML+SMU group relative to the contralateral muscles. This was unsurprising, as decreased fiber length is suspected to occur following VML injury and decreasing the number of sarcomeres in series decreases the optimal length of the muscle as a whole [3, 44]. 

The reported effects of denervation on muscle structure and function are numerous, and include muscle atrophy, sarcomeric disruption, and loss of specific force [41, 49, 50]. Evidence of denervation includes a shift in fiber type in that both fast and slow fibers to revert to a slow fiber type [45, 46]. Indeed, we observed a non-significant increase in the abundance and total area made up by type I fibers in injured muscles relative to the contralateral. The shift towards a greater abundance of slow fibers could have resulted in reduced force capabilities by contributing to the reduction in MF20+ cross-sectional area, as type I fibers are typically smaller in diameter than type II fibers. Furthermore, type I fiber grouping is well-documented and almost universally accepted as a sign of denervation and subsequent reinnervation [47, 48] and was observed in 70% of the animals in the 6-month recovery group. There was also as a significant negative correlation between the number of grouped fibers and the percentage of force recovery in the VML Only group; therefore, concomitant peripheral nerve injuries resulted in denervation which contributed to the total force deficits experienced in both experimental groups.”

Minor:

1. There were references that did not format correctly, lines 421 and 606.

Response: Thank you for pointing this out. They have been corrected. 

2. Please note the sex of the experimental animals, line 190.

Response: This has been modified to read: “Animals used for the surgical implant procedures were 6-7-month-old Polypay wethers (castrated males) weighing 45-55kg.” (lines 190-191).

3. Please add the absolute weight of the removed muscle tissue, line 338.

Response: The following has been added: “The absolute mass of skeletal muscle removed was 0.28 ± 0.06g and 0.34 ± 0.06g in the VML Only and the VML+SMU groups, respectively.” (lines 368-369).

4. In line 48, please consider removing “degenerative” as those types of disorders often have a different etiology than traumatic VML.

Response: This has been corrected to read: “Volumetric muscle loss is the traumatic or surgical loss of 20-30% or more of muscle volume.” (lines 47-48).

5. Since this is a new model, it would be useful to the reader to include force tracings of all experimental groups.

Response: The following figure has been included in the manuscript as Supplemental Figure 1.

S1 Fig 1. Representative Force Tracings. These images depict representative force tracings in response to a tetanic electrical stimulus in the contralateral ZM (A,D,G), the VML Only group (B,E,H), and the VML+SMU group (C,F,I) as well as animals in the 2-month (A-C), 4-month (D-F), and 6-month (G-I) recovery timepoints.

6. What was the cut off to consider a muscle fiber “small”, line 263 and 523? Was this uniformly set in FIJI? In other large animals studies less than 500um2 is often used.

Response: We apologize for the confusion. We have clarified the methods to explain this process. These fibers were identified based on their location within the tissue, not on their size, so “small” has been removed as a descriptor. Fibers in the repair site were identified by the technician who then quantified and measured the cross-sectional area of the fibers. This was clarified in the manuscript, lines 266-270.

“These same images were also used to evaluate the muscle fibers present in the injury site. These fibers were identified based on their location in the tissue rather than their size. The number and cross-sectional area of all of the fibers in the injury site was measured using ImageJ/Fiji in the 6-month recovery group animals.”

7. Figure 2, please clarify what the red data points are in panel C.

Response: This has been clarified in the manuscript (“...(denoted by the red data points in Fig 2C).”) as well as in the figure legend (“Red data points indicate the SMUs whose force peaked below the maximum current and/or frequency allowed by our force testing system.”). (lines 352 and 332-334, respectively.)

8. Figure 4, it could help readability if the time points were labeled near the A,B,C on the figure.

Response: We thank the reviewer for this suggestion. The figure has been updated to reflect this change. 

9. Line 477, please clarify is this muscle fiber cross sectional area? Or area of MF20 positive fibers?

Response: This has been clarified to read, “Combining all timepoints, there was no significant difference in total MF20+ area between the experimental groups (two-way RM ANOVA: P=0.7481)...” (line 511-513).

10. Line 533, was there any attempt to quantify how many centrally located nuclei there were?

Response: We did not attempt to quantify the number of centrally located nuclei. We did not believe that quantifying the central nuclei in one section would give an accurate representation of the muscle as a whole, as some fibers may have had central nuclei that were simply not visible in that section. 

Reviewer #2: In the article, titled “A tissue engineering approach for repairing craniofacial volumetric muscle loss in a sheep following a 2, 4, and 6-month recovery”, the authors generate one of the first craniofacial VML defect models in sheep and investigate the ability of skeletal muscle units (SMUs) to enhance recovery. The craniofacial VML model was demonstrated to generate complex injuries including vascular and neural damage, which confounded results investigating the contribution of SMUs to the repairing muscle. Overall, the study is of significant interest to the skeletal muscle community and speaks to issues regarding injury scaling and the treatment of craniofacial muscles. However, there are some concerns with data presentation and analysis that undermine the conclusions presented by the authors.

Methods:

Comment 1. Do the authors have data from age-matched control animals? In using the contralateral muscle as a control, there are potential issues with muscle hypertrophy or compensation via overuse of the contralateral. This could further obfuscate the contribution of SMUs to muscle repair. Commenting on this in the discussion may also be appropriate.

Response: During the post-operative monitoring period, we did not notice any change in motion that would indicate compensation by the contralateral ZM. We did not use age-matched control animals in this study, but we agree with the reviewer’s comment that a sham group or some sort of functional benchmarking prior to surgery would have been beneficial to address the possibility of compensation in the contralateral ZM.

The post-operative observations were clarified in the results: “None of the animals were observed to have abnormal eating habits after surgery nor were they observed to have any change in motion that would suggest compensation in the contralateral ZM.” (lines 403-405).

The following was added to the discussion: “There is also the possibility that the force deficits were exacerbated as the result of compensation of the contralateral ZM which would have resulted in its hypertrophy and increased the discrepancy in force production between the contralateral and injured ZMs. Although we did not observe any changes in motion that would have suggested compensation by the contralateral ZM, functional benchmarking prior to surgery and/or the use of sham animals will be taken into account in future studies.” (lines 805-810). 

Comment 2. How many slides/sections per animal were used to obtain IHC results (must use multiple sections per analysis)?

Response: One section of the entire midsubstance cross-section of the muscle was evaluated per animal for each histological analysis. Multiple sections were taken from the ZM midbelly from each animal, but appeared relatively identical, so one section was used for each histological analysis.

This was clarified in the manuscript in lines 260, 288, and 304.

Comment 3. Page 16, line 332: Mere presence of DAPI-stained nuclei does not eliminate the possibility of a necrotic core. The authors should present higher magnification images of the central regions of SMUs to assess nuclear morphology to claim lack of cell necrosis.

Response: We have included higher magnification images in Figure 2 I,J so that it is easier for the reader to visualize the nuclear morphology. This has been described in the manuscript: “Immunohistochemical staining also revealed the absence of a necrotic core, as DAPI-stained nuclei were present throughout the full thickness of the construct and exhibited normal nuclear morphology (Fig 2I,J).” (lines 359-361).

Comment 4. Page 17, lines 345-48: What was the rationale for implanting larger constructs into a single treatment group, as opposed to spreading them out across time points? Did “sentinel” SMUs from this group show differences in morphology? Without knowing more about these groups, it is unclear whether VML+SMU groups rescued patients/muscles from otherwise more severe injuries or not.

Response: The reason that all of the larger SMUs were implanted into the 6-month recovery group had to do with the schedule of animal surgeries. Because the 6-month animals had the longest recovery time, all of those animals underwent surgery first so that the entire experiment could be completed in a timely manner. At the time of SMU fabrication and subsequent implantation into the 6-month recovery group, we could not have anticipated that the SMUs fabricated at later times, with cells that had been frozen longer, would have grown more slowly and not developed that second monolayer. 

The sentinel SMUs fabricated as part of this cohort did not appear to exhibit any differences in morphology or quality of the monolayer compared to the other cohorts, other than increased size.

This was described in the results, surgical procedures section, lines 393-400: “The difference in SMU size was due to faster cell growth such that after the monolayer delaminated to form an SMU, a subsequent monolayer grew and delaminated around the existing SMU by the time of implantation. Because the 6mo. cohort of SMUs was fabricated first, the cells spent the least amount of time frozen prior to SMU fabrication which likely accounts for the difference in the cells’ growth rate. It is important to note that although larger, these SMUs did not exhibit any differences in monolayer quality, morphology, or force production compared to the other sentinel SMUs.” 

Comment 5. Along the same line as above, did the larger SMUs have increased force production (or specific force)?

Response: As described in the manuscript, unfortunately we were not able to collect data on the true maximum force production of the SMUs, as only n=2 SMUs achieved maximum force before we maxed out the current and/or frequency able to be supplied by our SMU force testing system (described in lines 350-354). There were 3 sentinel SMUs fabricated from the cohort that was implanted into the 6-month animals. Of these, one SMU did achieve its maximum force (one of the red dots in Fig2C, force = 135uN). However, we were not able to measure the true maximum force of the other 2 SMUs (force = 36.9 and 34.2uN). Because these are comparable to forces achieved by the SMUs in other cohorts, we cannot say that these larger sentinel SMUs had increased force production over the others. 

Comment 6. Page 20, line 421: Please correct/remove this error message, and supply the intended text.

Response: We apologize for this. We hope that there is no error message in this new version.

Comment 7. Page 29, line 606: Please correct/remove this error message, and supply the intended text.

Response: We apologize for this. We hope that there is no error message in this new version.

Comment 8. Did the authors see histologic evidence of the SMUs at any time point? The authors’ previous work has shown SMU integration with the host, but usually they are distinguishable from host muscle. If the SMUs are not present, do the authors have a hypothesis as to why?

Response: We have noted the presence of small muscle fibers in the injury site in all recovery time points and in both the VML and VML+SMU group (described in lines 638-651). Because small fibers are present in the VML only group (having no tissue implanted), we know at least some of these small fibers are atrophied and/or regenerating fibers. It is possible that some of the small fibers present in the VML+SMU group were donated by the SMU but we cannot definitively identify the origin of the fibers. The longer recovery timepoints and greater degree of remodeling in the sheep VML model compared to the rat VML model has prevented us from being able to definitively identify the SMU. In a previous rat study, we have used fluorescently labeled cells to track the integration of the SMU with the host, however this was not possible with sheep. In future studies, we hope to label the SMU cells to be able to definitively identify the SMU within the repair site.

Comment 9. In conjunction with the analysis of fat content in the muscles (page 29, lines 605-616), did the amount of collagen (i.e. scar formation) change between treatment groups (i.e. quantification of Fig 7D-F)?

Response: We acknowledge that a measure of fibrosis is important, so we measured the cross-sectional area of the muscle (including the fibrotic region) and subtracted the MF20+ cross-sectional area to get the MF20- cross-sectional area. The MF20- area includes the fibrotic connective tissue but will also include any blood vessels or nerves associated with the tissue. 

The following has been added to the methods, lines 273-286: 

“To quantify the total cross-sectional area of the muscle, including both MF20+ muscle fibers as well as connective tissue, nerve, and blood vessels within the muscle, the same MF20 and laminin-stained midbelly ZM cross-sections were used. The total cross-sectional area was measured in ImageJ/Fiji using the freehand selection tool. This total cross-sectional area was then used to calculate the total cross-sectional area specific force (N/cm2) by dividing the maximum tetanic force of the muscle by its total cross-sectional area. The comparison of the MF20+ specific force and the total cross-sectional area specific force gives insight into the change in muscle content, including the increase in fibrotic connective tissue in the injured muscles. 

 Finally, to quantify the difference in the amount of connective tissue between the uninjured contralateral muscles and the injured ZMs, we calculated the MF20- area by subtracting the MF20+ area from the total cross-sectional area and represented it as a percentage of the total cross-sectional area. This portion of the total area that was made up of connective tissue was then normalized to the amount that was present in the contralateral muscle.”

This has been added to the results, lines 559-616: 

“We calculated the total cross-sectional area of the injured and uninjured ZMs to evaluate whether there were changes in the muscle size as a result of the VML injury. We normalized the total cross-sectional area of the surgical ZM to the contralateral and found that there was no significant difference in normalized MF20+ area between recovery timepoints (P=0.2258) or between experimental groups (P=0.6753). The average normalized total cross-sectional area was 74.0 ± 27.2% in the VML Only group (n=11) and 79.6 ± 15.1% in the VML+SMU group (n=12) (Fig 7A). Combining all timepoints, there was no significant difference in total cross-sectional area between the experimental groups (two-way RM ANOVA: P=0.5769), but there was a significant difference between the contralateral and surgical ZMs (two-way RM ANOVA: P<0.0001) (Fig 7B). Using a Sidak’s multiple comparisons test, there was a significant difference between the total cross-sectional area in the contralateral and surgical ZMs of both experimental groups (P=0.0008, n=11 for VML Only; P=0.0070, n=12 for VML+SMU) indicating loss of ZM cross-sectional area in both experimental groups at the time of explant with respect to the uninjured contralateral.

The specific force calculated by using the MF20+ cross-sectional area revealed that there was no significant difference between the injured and uninjured ZMs which indicates that the remaining muscle tissue was healthy. However, calculating specific force using the total cross-sectional area of the muscle, which includes the muscle fibers as well as connective tissue, reveals the overall health of the ZM. Using the total cross-sectional area specific force and normalizing it to the contralateral, the mean normalized total cross-sectional area specific force was 74.8 ± 32.5% in the VML Only group (n=11) and 77.5 ± 27.6% in the VML+SMU group (n=12) (Fig 7C). A two-way ANOVA showed that there was no significant difference in the total-cross sectional area specific force as a percentage of the contralateral between experimental groups (P=0.6753). Combining all timepoints, there was no significant difference in the total cross-sectional area specific force between experimental groups (P=0.2594) but there was a difference between the contralateral and surgical ZMs (P=0.0009) using a paired two-way ANOVA. Notably, there was a significant difference in the specific force between the surgical ZM and the uninjured contralateral ZM in both the VML Only (P=0.0384, n=11) and the VML+SMU (P=0.0165, n=12) group (Fig 7D). This indicates that the functional capacity of the ZM as a whole was decreased as a result of the VML injury. 

To quantify the amount of connective tissue present in the ZM, that is MF20- tissue, we subtracted the MF20+ area from the total cross-sectional area and represented this as a percentage of the total area. When normalizing the percentage of MF20- tissue to the control, a two-way ANOVA revealed that there was no significant difference between recovery timepoints (P=0.1138) or between experimental groups (P=0.1016). Specifically, the MF20- area was 156 ± 48.4% of the contralateral in the VML Only group and 187 ± 53.7% of the contralateral in the VML+SMU group, indicating an over 50% increase in connective tissue in the injured ZMs (Fig 7E). Combining all recovery timepoints, there was no significant difference between experimental groups (P=0.6133) but there was a significant difference between the contralateral and surgical ZMs (P<0.0001). Specifically, the percentage of the cross-sectional area that was MF20- was 26.3 ± 5.5%, 41.7 ± 12.9%, 46.3 ± 12.0% in the contralateral, VML Only, and VML+SMU groups, respectively (Fig 7F). Overall, this indicates a significant increase in the connective tissue content of the injured ZMs of both experimental groups at all recovery timepoints.”

The additional figure and figure legend, for your reference:

Fig 7. Total Cross-Sectional Area and Specific Force. (A) A two-way ANOVA revealed that there was no significant difference in the total cross-sectional area as a percentage of the contralateral between experimental groups (P=0.6753). (B) In both experimental groups, the total cross-sectional area of the surgical ZM was significantly lower than the uninjured contralateral (P=0.0008, n=11 for VML Only; P=0.0070, n=12 for VML+SMU). (C) There was no significant difference in normalized total cross-sectional area specific force between experimental groups (P=0.0194). (D) In both experimental groups, the total cross-sectional area specific force was significantly lower in the surgical ZM than the uninjured contralateral (P=0.0384, n=11 for VML Only; P=0.0165, n=12 for VML+SMU). (E) There was no significant difference in the MF20- area as a percentage of the contralateral between recovery timepoints (P=0.1138) or between experimental groups (P=0.101). (F) In both experimental groups, the MF20- area as a percentage of the total cross-sectional area was significantly higher than the uninjured contralateral (P=0.0007, n=11 for VML Only; P<0.0001, n=12 for VML+SMU).

Comment 10. Did the authors note any differences in muscle repair/phenotype as a function of depth within the tissue?

Response: We did not observe any differences in muscle repair/phenotype as a function of depth within the tissue. However, in all animals, there was a thick layer of scar tissue on the superficial side of the muscle, tethering the muscle to the skin. Histologically, in cross-section beneath that layer, there was no gradation of muscle repair or phenotype with respect to tissue depth. 

This was further described in the results, gross observations at explant section, “This layer was thickest on the superficial side of the muscle, tethering it to the skin.” (lines 431-432).

Discussion:

Comment 11. Page 30, line 628: Sentence ends with hanging clause “…reasons.” Please finish the sentence, copying from the introduction if necessary. 

Response: Thank you for pointing out that typo. The clause was corrected to say “a variety of reasons” with the reasons being identified in the following sentence. 

Comment 12. Page 31, line 641: It is not wholly clear why the benchmark of 70% was selected to be a comparison of note. Likely, the rationale is that this percentage would indicate a simple “increase” in the force production beyond a theoretical loss of 30% volume. However, the authors, as well as the field, acknowledge that this assumption of a linear loss of function with respect to defect size is incorrect and not valid. The authors should omit this comparison, or focus on the patients with ~100% force production with respect to the contralateral.

Response: We have modified the discussion and removed the 70% benchmark as a point of comparison. 

Comment 13. Page 32, lines 668-671: The authors make an interesting point regarding the role of inflammation in the repair of VML defects. Did the authors assess (qualitatively or quantitatively) inflammation within these sites? An assessment of chronic inflammation may strengthen these claims.

Response: We did not quantitatively assess inflammation, but we agree that this would be a beneficial assessment to include in the future. Our bloodwork only sought to evaluate immune response governed by white blood cells, but it would have been useful to measure erythrocyte sedimentation rate or C-reactive protein, among others, as measures of systemic inflammation. Measures of acute inflammation would also give better insight into the inflammatory mechanisms of VML and the potential modulation provided by SMU implantation. This is a great suggestion and will be taken into account in future studies.

Comment 14. Page 32, line 672: Reference 24 removed 5.26 +/- 0.76 g of the peroneus tertius muscle, whereas this study removed ~ 0.6-0.72 g (exact mass not reported) of the zygomaticus major muscle. These are not “identical”. While the intent of the authors might be that the percentage of tissue that was removed was similar (~30%), the volume is an order of magnitude different. While this may further support the authors narrative (i.e. craniofacial VML is more complex), the authors should revise this paragraph to more concisely make their comparisons and derive their conclusions.

Response: The exact mass of muscle removed has been added to the manuscript: “The absolute mass of skeletal muscle removed was 0.28 ± 0.06g and 0.34 ± 0.06g in the VML Only and the VML+SMU groups, respectively.” (lines 368-369).

The paragraph has been rewritten as follows: “The geometry and location of the VML injury induced in this study may have created comorbidities that increased the severity of the injury and contributed to variability within groups. Specifically, as is common in extremity models, we chose to dissect a full thickness longitudinal portion of the ZM constituting 30% of the muscle mass (Fig 1A). In the native ZM, the vasculature inserts on the deep side of the muscle which made it difficult to create the VML injury without damaging the vascular bundle. Indeed, despite our attempts to avoid injuring native vasculature, in some instances, bleeding occurred during the muscle removal. We believe this injury led to ischemia that caused coagulative necrosis of the muscle in n=5/30 animals, resulting in virtually no muscle fibers present in the midbelly of the ZM at the time of explant. It is well understood that ischemia leads to muscle necrosis, and total muscle necrosis can occur if ischemia occurs for prolonged periods of time [40, 52]. Thus, the variability within groups may be related to variability in the severity of damage to the native vasculature. Moreover, if vascular damage led to necrosis of the remaining native muscle, it could have prevented survival of the SMU as well, as viability of implanted engineered tissues is dependent on rapid vascularization of the construct by the surrounding tissue [53].”

Comment 15. The authors posit that ischemic conditions may also be affecting the functional recovery of the injuries. Did they quantify the amount of blood vessels in the wound sites and if so, were differences observed?

Response: Although we did attempt to identify the blood vessels in the injury site, the growth of the vessels throughout the injury site did not occur in an organized manner and would have been impossible to accurately quantify. We agree that this would be a useful assessment to include in future studies; perhaps we could quantify tissue perfusion.

Figures

Comment 16. Fig 1D – Did the authors stitch the remaining muscle together with the SMUs after the generation of the VML – is that what this image is showing? If so, was this similarly done (or why not done) in the VML only condition?

Response: The remaining muscle in the VML Only animals was not stitched together. This was to simulate a true negative control without any interventions taken to repair it. This was clarified in the methods, lines 212-214: “The VML Only animals (negative control) received the injury and neurotization without any additional repair; the remaining muscle was not sutured together so as to simulate a true negative control (Fig 1B).”

Comment 17. Fig 2B – Please increase contrast/visibility of myofibers in this image.

Response: Here is the image with the contrast adjusted, for your reference. The figure has been updated to reflect this change.

Comment 18. Fig 2F/G – The text states that n=2 SMUs were used in implantation studies, but there appear to be 3 discrete “bundles” in these images. Are there 3 SMUs?

Response: We understand the image depicting the modular SMU (Fig2 F,G) may be confusing. This modular SMU, while there appears to be 3 bundles, is only 2 individual tissue units. Because of the nature of the SMUs (a rolled-up and fused monolayer), the degree of fusion of the monolayer affects its shape in cross-section. For example, the image below is of one SMU in cross-section, with the monolayer having rolled up on each side to meet in the middle. You can see in the image that the tissue was connected at the bottom, although sectioning artifact has disrupted it in this example. Thus, the fusion of the monolayers/tissue units may have occurred unevenly to create the shape of the modular SMU in Fig2 F,G.

Comment 19. Fig 9 – How can you parse the expression of synaptic vesicle protein-2 against neurofilament, when they are visualized with the same fluorophore? Are some images visualizing one marker over the other? In general, the quality of these images could be improved. Interpretation of what are NMJs is unclear, the authors could indicate representative NMJs in the images, as this becomes important within the discussion (page 32, lines 665-668)

Response: While they are visualized with the same fluorophore, the junctions would not be fully formed (i.e. innervated) unless there was overlap of the red and the green structures, creating a yellow region. We observed overlap of these structures in all of the contralateral animals and in specific instances in the recovery groups. 

We agree, this method of identifying NMJs is purely qualitative and meant to provide visual support of the denervation hypothesis. We have clarified the results and discussion to better explain our interpretation of the NMJ histology.

In the results: “We also immunohistochemically stained for the presence of neuromuscular junctions (NMJs) in longitudinal sections of the injury site and in the uninjured contralateral muscles (Fig 9). An NMJ was noted to be present if there was an overlap of the pre-synaptic and post-synaptic structures which resulted in a yellow region due to the overlap of the red and green fluorophores. This overlapping of structures was observed in the contralateral ZM at all recovery timepoints, and at least one NMJ was identified in the repair site of both the VML Only and the VML+SMU groups at the 6mo. timepoint. Although acetyl choline receptors and neurofilament were present at earlier timepoints, small muscle fibers in the injury site do not appear to be reinnervated (no overlapping structures) in the 2mo. and 4mo. recovery groups.” (lines 658-666)

In the discussion: “Furthermore, we did not see evidence of any fully formed neuromuscular junctions in the repair site at the 2mo. or 4mo. recovery timepoints, but we did note at least one NMJ at the 6mo. recovery point in both experimental groups. However, even at 6 months, the presence of NMJs in the repair site was limited, suggesting that a recovery period greater than 6-months is necessary to obtain functional reinnervation in this facial VML model.” (lines 826-830)

Comment 20. Fig 12 – Please increase the contrast of images. While the reviewer appreciates the low percentage of fat tissue present in the samples, the quality of these images could be improved.

Response: Below is the modified figure, for your reference.

---

## [Decision Letter · Decision Letter 1]

19 Aug 2020

PONE-D-20-11394R1

A Tissue Engineering Approach for Repairing Craniofacial Volumetric Muscle Loss in a Sheep Following a 2, 4, and 6-Month Recovery

PLOS ONE

Dear Dr. Larkin,

Thank you for submitting your manuscript to PLOS ONE. After careful consideration, we feel that it has merit but does not fully meet PLOS ONE’s publication criteria as it currently stands. Therefore, we invite you to submit a revised version of the manuscript that addresses the points raised during the review process.

We look forward to receiving your revised manuscript.

Kind regards,

Warren Grayson

Academic Editor

PLOS ONE

Reviewers' comments:

Reviewer's Responses to Questions

**Comments to the Author**

1. If the authors have adequately addressed your comments raised in a previous round of review and you feel that this manuscript is now acceptable for publication, you may indicate that here to bypass the “Comments to the Author” section, enter your conflict of interest statement in the “Confidential to Editor” section, and submit your "Accept" recommendation.

Reviewer #1: (No Response)

Reviewer #2: All comments have been addressed

2. Is the manuscript technically sound, and do the data support the conclusions?

Reviewer #1: Partly

Reviewer #2: Yes

3. Has the statistical analysis been performed appropriately and rigorously? 

Reviewer #1: Yes

Reviewer #2: Yes

4. Have the authors made all data underlying the findings in their manuscript fully available?

Reviewer #1: Yes

Reviewer #2: Yes

5. Is the manuscript presented in an intelligible fashion and written in standard English?

Reviewer #1: Yes

Reviewer #2: Yes

6. Review Comments to the Author

Reviewer #1: In general the authors adequately responded to most of my previous queries. Two points remain from my original review that need to be clarified.

1) Regarding figure 2C, the authors need to clarify how there was a value obtained between the two peak forces that were above the range of testing. Also it would be more appropriate to average the SMU forces for those values that are acceptable with the exclusion of the two points. It is not clear why these were averaged in, as the authors cannot say if they were 100uN or 150uN, thus averaging them as 100 is not correct even if it is stated as an underestimation.

2) Regarding the NMJs, it is still unclear why so many of the images for Figure 9 are cross-sections. The claim of disorganized fibers is fine but not well supported by other figures in this manuscript. This does not provide much confidence in this analysis or evaluation, to this reviewer. It would seem the authors are providing the best image they can to make a point and not one that is supportive of the data as a whole.

The response notes these qualitative analysis was “meant to provide visual support of the denervation hypothesis” however the manuscript states that NMJs were noted as present if there was overlap of pre and post synaptic structures, which would not support analysis of denervation. It only supports the identification of innervated NMJs. Further, how many total NMJs were identified (fully formed or not)? In my original comment I suggested the discussion on this should be toned down and it was really not. The authors do not provide data that support denervation. Including the qualitative analysis and figures is completely acceptable, but it needs to be discussed as such. Conversely, if the authors are going to keep in the discussion points they made they should note how many NMJs per muscle were evaluated. If they did evaluate the number of denervated (non-overlapping) and innervated (overlapping) NMJs. Keeping in mind that there are 10,000-20,000 fibers (and subsequent NMJs) per muscle based on the data they provided. Thus, if the authors are only providing evidence of the perhaps 3 NMJs per muscle indicated in figure 9 the comments should be clear.

Again the data provided do not support the discussion of denervation and the authors should rework the discussion of the data directly provided.

3) Regarding clustering and denervation link, there are a few statements in the discussion that are not clear. On line 795 (marked version) the authors make clear there is a non-significant change but the authors go on to discuss the implication of this. Additionally, the last sentence of this paragraph (lines 801-804) is unclear. The statement in lines 819 and 850 are not supported by the data and should be edited.

Reviewer #2: (No Response)

7. PLOS authors have the option to publish the peer review history of their article (what does this mean?). If published, this will include your full peer review and any attached files.

Reviewer #1: No

Reviewer #2: No

---

## [Author Response · Author response to Decision Letter 1]

26 Aug 2020

PONE - D-20-11394R1

Review Comments to the Author

Reviewer #1: In general, the authors adequately responded to most of my previous queries. Two points remain from my original review that need to be clarified.

Comment: Regarding figure 2C, the authors need to clarify how there was a value obtained between the two peak forces that were above the range of testing. 

Response: To clarify, the equipment responsible for producing the electrical stimulus was what maxed out, not the equipment responsible for measuring the force of the contractions. Our current stimulator can only elicit a maximum current of 1000mA and a frequency of 120Hz. At this current and/or frequency, and due to the size of these large constructs, we were unable to reach maximum tetanic force capabilities of the SMU constructs. Therefore, using this equipment, we maxed out the capabilities of our equipment and were only able to elicit maximum tetanic force measures in two constructs (shown as red points in Figure 2C). So, the maximum forces capabilities of all but two of the SMUs are underestimated in this study. We have added this statement to the methods (Page 9, line 180) and results (Page17, Line 357) section of the paper. We feel that maximum forces that we could elicit with our current equipment is valuable to the reader. 

Comment: Also, it would be more appropriate to average the SMU forces for those values that are acceptable with the exclusion of the two points. 

Response: It is actually only on two points that we were able to reach the peak of force production, the rest of the SMUs maxed out the equipment before reaching peak. So, we could just present those two data points as an example of potential maximum tetanic force for the SMUs and eliminate all of the data points that maxed out the equipment.

Comment: Regarding the NMJs, it is still unclear why so many of the images for Figure 9 are cross-sections. 

Response: I am sorry for the confusion. However, the NMJ’s are in Figure 10 and are longitudinal sections, specifically of the repair site. Adjacent to the repair site is native muscle fibers that are indeed oriented longitudinally. However, because the repair site is characterized primarily of disorganized scar tissue, the small muscle fibers present there grew disorganized and are not oriented longitudinally with the rest of the muscle. For this reason, a longitudinal section of the ZM muscle as a whole revealed some fibers and NMJs in the repair site that appeared to be in cross-section. From the paper: “Immunostaining of longitudinal sections for acetylcholine receptors (red), synaptic vesicle protein-2 (green), and neurofilament (green) was performed to identify the presence of neuromuscular junctions.”

Comment: The claim of disorganized fibers is fine but not well supported by other figures in this manuscript. This does not provide much confidence in this analysis or evaluation, to this reviewer. It would seem the authors are providing the best image they can to make a point and not one that is supportive of the data as a whole.

Response: There is a great abundance of scar tissue in the repair site, as noted in figures 8 and 11; however, these are zoomed out images of the entire ZM and from these, the reader is not be able to appreciate the morphology of the repair site. Figures 9 and 10 do show high magnification regions of the repair site and also demonstrate the cross-sectional versus longitudinal phenomenon: even though the images are cross-sections, the disorganized nature of the repair site result in some fibers presenting as longitudinal (see Fig 9D, E,F). The same is true for Fig10 (NMJs) in which a longitudinal section of the muscle as a whole result in some fibers appearing cross-sectional. 

Comment: The response notes these qualitative analyses was “meant to provide visual support of the denervation hypothesis” however the manuscript states that NMJs were noted as present if there was overlap of pre and post synaptic structures, which would not support analysis of denervation. It only supports the identification of innervated NMJs. Further, how many total NMJs were identified (fully formed or not)? 

Response: We apologize for the confusion. These were the only NMJs that were able to identified within the repair site. The results specified that “at least one NMJ” was able to be identified. This was supposed to highlight the lack of abundance of NMJs in the repair site. The point being made was that there was a stark absence of NMJs in the repair site, suggesting that the repair site is widely denervated. This was clarified in the results, page 31, line 671 and 675: “However, NMJs were widely absent in the majority of the repair site of these experimental groups and the images shown in Fig 10 depict what were usually the only NMJs present. Additionally, although acetyl choline receptors and neurofilament were present at earlier timepoints, small muscle fibers in the injury site do not appear to be reinnervated (no overlapping structures) in the 2mo. and 4mo. recovery groups. Overall, in all experimental groups at all timepoints, there appears to be a lack of innervation in the repair site.”

This was also described in the discussion: “However, even at 6 months, the presence of NMJs in the repair site was very limited, suggesting that a recovery period greater than 6-months is necessary to obtain functional reinnervation of the repair site in this facial VML model.” Again, the point being made was that innervation in the repair site was lacking, both in quantity (only several NMJs identified in the entire recovery site) and quality (the fibers are disorganized and there is no overlap of the pre and post synaptic structures in the 2 or 4 mo. groups).

In my original comment I suggested the discussion on this should be toned down and it was really not. The authors do not provide data that support denervation. Including the qualitative analysis and figures is completely acceptable, but it needs to be discussed as such. Conversely, if the authors are going to keep in the discussion points, they made they should note how many NMJs per muscle were evaluated. If they did evaluate the number of denervated (non-overlapping) and innervated (overlapping) NMJs. Keeping in mind that there are 10,000-20,000 fibers (and subsequent NMJs) per muscle based on the data they provided. Thus, if the authors are only providing evidence of the perhaps 3 NMJs per muscle indicated in figure 9 the comments should be clear.

Response: Although there are many muscle fibers in the ZM, we only sought to identify the presence of NMJs in the repair site (with the exception of the contralateral muscle). NMJs overall were widely absent in the repair site. We did not see any NMJs in the 2 or 4mo. groups (specified in the manuscript already) and we only saw several NMJs in the entire repair site of the animals in the 6mo. group. This was specified in the results (see response above for excerpt).

Again, the data provided do not support the discussion of denervation and the authors should rework the discussion of the data directly provided.

Response: The qualitative image of the NMJ does not and was not meant to support the argument for denervation. It was solely meant to describe the repair site as lacking innervation at the time of explant. But the fiber grouping results definitely point to denervation and reinnervation. The number of group fibers is quantified (and correlated with percentage of force recovery) in Fig 12. From the figure, you can see that the number of groups fibers present ranged from 8 to 282, and as specified in the manuscript, 70% of the animals had more than 50 grouped fibers. 

From the discussion: “The location of our VML injury relative to the MEPs suggests that the muscle resection potentially created a denervation injury in the remaining muscle. This is corroborated by the notable type I fiber grouping, an indication of denervation and subsequent reinnervation [47, 48], that was observed in n=7/10 animals in the 6mo. recovery group.”

Regarding the abundance of type I fibers (Fig 11D), although as a group there is no significant difference from the contralateral, it is apparent that 70% of the animals experienced a large shift in the number of slow fibers, while only 3 animals had a quantity of slow fibers that was within the range of the contralateral. While this were not significant, there is definitely evidence for denervation in these n=7/10 animals, which is also supported by the prevalent type I fiber grouping in these same animals. This line was added to the discussion, line 807, “However, 70% of the animals had a quantity of type I fibers that exceeded the highest number exhibited by the contralateral.” 

3) Regarding clustering and denervation link, there are a few statements in the discussion that are not clear. On line 795 (marked version) the authors make clear there is a non-significant change but the authors go on to discuss the implication of this. Additionally, the last sentence of this paragraph (lines 801-804) is unclear. The statement in lines 819 and 850 are not supported by the data and should be edited.

Response: We have gone through the entire manuscript and better defined the fiber grouping results as an indication of acute denervation due to the injury and subsequent reinnervation. 

Regarding line 795: Although as a group, this shift is not significant, 70% of the animals exhibited a quantity of type I fibers that exceeded the maximum number exhibited by the contralateral. The lack of significance does not suggest there is not a shift in the number of type I fibers, but rather that the shift was not experienced equally by all animals. Much of our data suggests that the animals experienced a differential severity of the initial VML injury, and this is such evidence. The fiber grouping (more than 50 grouped fibers present in 70% of the animals) provides concrete evidence of denervation and subsequent reinnervation in those animals. 

The sentence in lines 801-804 was clarified: “There was also as a significant negative correlation between the number of grouped fibers and the percentage of force recovery in the VML Only group such that a larger number of grouped fibers was significantly correlated with a lower percentage of force recovery. This suggests that the denervation evidenced by the notable fiber grouping likely contributed to the total force deficits experienced in the VML Only groups.”

Response: Line 814 - 818: We reworded this section for clarity.

Response: Line 864: The line was edited to read: “Additionally, the results showed evidence of denervation.” We believe that we have presented sufficient evidence of denervation. 

Reviewer #2: (No Response)

---

## [Decision Letter · Decision Letter 2]

1 Sep 2020

A Tissue Engineering Approach for Repairing Craniofacial Volumetric Muscle Loss in a Sheep Following a 2, 4, and 6-Month Recovery

PONE-D-20-11394R2

Dear Dr. Larkin,

We’re pleased to inform you that your manuscript has been judged scientifically suitable for publication and will be formally accepted for publication once it meets all outstanding technical requirements.

Kind regards,

Warren Grayson

Academic Editor

PLOS ONE

Additional Editor Comments (optional):

Reviewers' comments:

Reviewer's Responses to Questions

**Comments to the Author**

1. If the authors have adequately addressed your comments raised in a previous round of review and you feel that this manuscript is now acceptable for publication, you may indicate that here to bypass the “Comments to the Author” section, enter your conflict of interest statement in the “Confidential to Editor” section, and submit your "Accept" recommendation.

Reviewer #1: All comments have been addressed

2. Is the manuscript technically sound, and do the data support the conclusions?

Reviewer #1: Yes

3. Has the statistical analysis been performed appropriately and rigorously? 

Reviewer #1: Yes

4. Have the authors made all data underlying the findings in their manuscript fully available?

Reviewer #1: Yes

5. Is the manuscript presented in an intelligible fashion and written in standard English?

Reviewer #1: Yes

6. Review Comments to the Author

Reviewer #1: (No Response)

7. PLOS authors have the option to publish the peer review history of their article (what does this mean?). If published, this will include your full peer review and any attached files.

Reviewer #1: No

---

## [Editor Report · Acceptance letter]

11 Sep 2020

PONE-D-20-11394R2 

A Tissue Engineering Approach for Repairing Craniofacial Volumetric Muscle Loss in a Sheep Following a 2, 4, and 6-Month Recovery 

Dear Dr. Larkin:

I'm pleased to inform you that your manuscript has been deemed suitable for publication in PLOS ONE. Congratulations! Your manuscript is now with our production department. 

Kind regards, 

on behalf of

Dr. Warren Grayson 

Academic Editor

PLOS ONE